# An image-computable model of human visual shape similarity

**Yaniv Morgenstern**[1]*, **Frieder Hartmann**[1], **Filipp Schmidt**[1], **Henning Tiedemann**[1], **Eugen Prokott**[1], **Guido Maiello**[1], **Roland W. Fleming**[1,2]

**1** Department of Experimental Psychology, Justus-Liebig University Giessen, Giessen, Germany, **2** Center for Mind, Brain and Behavior (CMBB), University of Marburg and Justus Liebig University Giessen, Giessen, Germany

* Yaniv.Morgenstern@psychol.uni-giessen.de

## Abstract

Shape is a defining feature of objects, and human observers can effortlessly compare shapes to determine how similar they are. Yet, to date, no image-computable model can predict how visually similar or different shapes appear. Such a model would be an invaluable tool for neuroscientists and could provide insights into computations underlying human shape perception. To address this need, we developed a model ('ShapeComp'), based on over 100 shape features (e.g., area, compactness, Fourier descriptors). When trained to capture the variance in a database of >25,000 animal silhouettes, ShapeComp accurately predicts human shape similarity judgments between pairs of shapes without fitting any parameters to human data. To test the model, we created carefully selected arrays of complex novel shapes using a Generative Adversarial Network trained on the animal silhouettes, which we presented to observers in a wide range of tasks. Our findings show that incorporating multiple ShapeComp dimensions facilitates the prediction of human shape similarity across a small number of shapes, and also captures much of the variance in the multiple arrangements of many shapes. ShapeComp outperforms both conventional pixel-based metrics and state-of-the-art convolutional neural networks, and can also be used to generate perceptually uniform stimulus sets, making it a powerful tool for investigating shape and object representations in the human brain.

## Author summary

The ability to describe and compare shapes is crucial in many scientific domains from visual object recognition to computational morphology and computer graphics. Across disciplines, considerable effort has been devoted to the study of shape and its influence on object recognition, yet an important stumbling block is the quantitative characterization of shape similarity. Here we develop a psychophysically validated model that takes as input an object's shape boundary and provides a high-dimensional output that can be used for predicting visual shape similarity. With this precise control of shape similarity, the model's description of shape is a powerful tool that can be used across the neurosciences and artificial intelligence to test role of shape in perception and the brain.

**Data Availability Statement:** The data can be accessed at https://doi.org/10.5281/zenodo.4730985.

**Funding:** This research was funded by the DFG funded Collaborative Research Center "Cardinal

Mechanisms of Perception" (222641018–SFB/TRR 135 TP C1) and the ERC Consolidator award "SHAPE" (ERC-CoG-2015-682859). G.M. was supported by a Marie-Skłodowska-Curie Actions Individual Fellowship (H2020-MSCA-IF-2017: 'VisualGrasping' Project ID: 793660). The funders had no role in study design, data collection and analysis, decision to publish, or preparation of the manuscript.

**Competing interests:** The authors have declared that no competing interests exist.

## Introduction

One of the most important goals for biological and artificial vision is the estimation and representation of shape. Shape is the most important cue in object recognition [1–4] and is also crucial for many other tasks, including inferring an object's material properties [5–9], causal history [10–13], or where and how to grasp it [14–18]. Here we focus on how the visual system determines the perceptual similarity between different shapes, which is thought to be a core stage in object perception [19–22] and often used to probe shape processing in the brain [23–26]. Shape is also central to many other disciplines, including computational morphology [27], anatomy [28], molecular biology [29], geology [30], meteorology [31], computer vision [32], and computer graphics [33]. For all these fields, it would be exceedingly useful to be able to characterize and quantify the visual similarity between different shapes automatically and objectively (**Fig 1A**).

Here we sought to develop and validate a model to estimate perceived 2D shape similarity, directly from images, by combining numerous shape metrics. Our goal was to implement into a concrete, executable, image-computable model, the widely-held notion that human visual similarity perception integrates multiple shape descriptors. Specifically, given a pair of shapes, $\{f_1, f_2\}$, the model should compare and combine shape metric $i$ (of a total of $N$) to predict the perceived similarity between shapes, $\hat{s}$, on a continuous scale (**Fig 1B**), $\hat{s} = \sqrt{\sum_{i=1}^{N} (f_{1i} - f_{2i})^2}$.

Although real-world objects are 3D, humans can make many inferences from 2D contours (e.g., [13, 34, 35]). Many 2D shape representations have been proposed—both for computational purposes and as models of human perception—each summarizing the shape boundary or its interior in different ways (**Fig 1B**; [32]). These include (but are not limited to) *basic shape descriptors* (e.g., area, perimeter, solidity; [36]), *local comparisons* (e.g., Euclidean distance; Intersection-over-Union, IoU; [37]), *correspondence-based metrics* (e.g., shape context; [38]), *curvature-based metrics* [39], *shape signatures* (see [32]), *shape skeletons* [40], and *Fourier descriptors* [41].

These different shape descriptors have complementary strengths and weaknesses. Each one is sensitive to certain aspects of shape, but relatively insensitive to others (**Fig 1C–1E**). For example, some metrics are entirely scale or rotation invariant, while others vary depending on the size or orientation of the object. We tested whether combining many complementary shape descriptors into a multidimensional composite would capture the many different ways that human observers compare shapes. We begin by analyzing a large database of real-world shapes and show that different descriptors do indeed tap into different aspects of shape.

### Complementary nature of different shape descriptors

To appreciate the complementary nature of different metrics—and the necessity of combining them—consider that human visual shape representation is subject to two competing constraints (See also, [42–44]). On the one hand, to achieve stable object recognition across changes in viewpoint and object pose, it is useful for shape descriptors to deliver consistent descriptions across large changes in the retinal image ('robustness'). On the other hand, to discriminate finely between different objects with similar shapes, shape descriptors must discern subtle changes in shape ('sensitivity'). These two goals are mutually exclusive and different descriptors necessarily represent a trade-off between them. Yet, the tradeoff for a given shape descriptor depends on which set of shape transformations we consider. This becomes evident when we organize descriptors along a continuum that describes their robustness to changes in shape across a range of transformations—such as rotation, scaling, shearing, or adding noise.

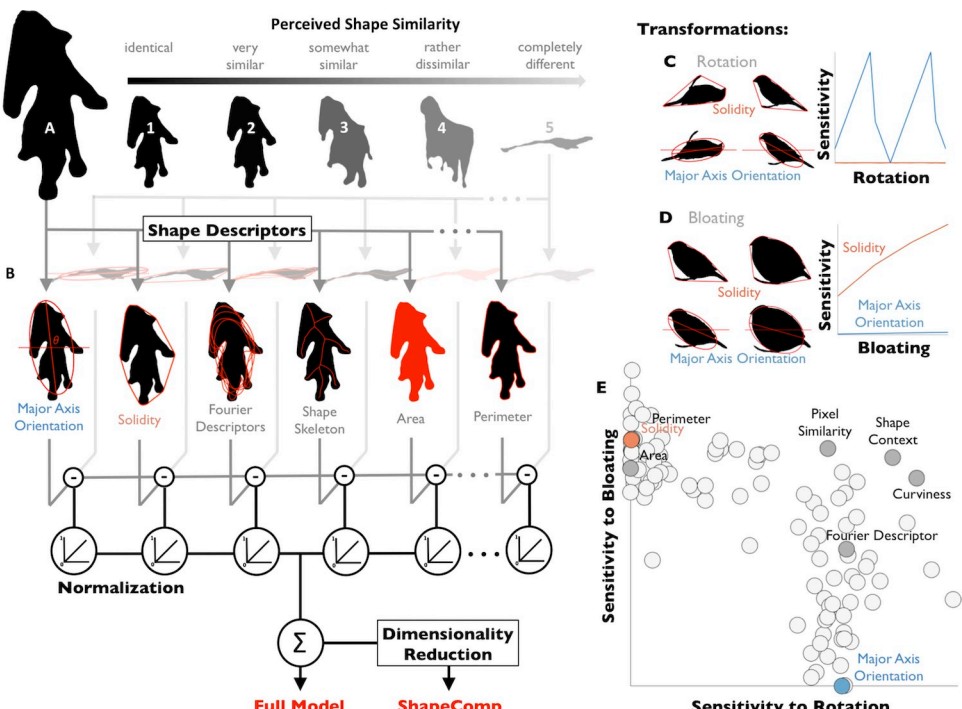

**Fig 1. ShapeComp: a multidimensional perceptual shape similarity model.** We readily perceive how similar shape (**A**) is from others (numbered 1–5). (**B**) Outline of our model, which compares shapes across >100 shape descriptors (6 examples depicted). The distance between shapes on each descriptor was scaled from 0 to 1 based on the range of values in a database of 25,712 animal shapes. Scaled differences are then linearly combined to yield 'Full Model' response. Applying MDS to >330 million shape pairs from the Full Model yields a multidimensional shape space for shape comparison ('ShapeComp'). We reasoned that many descriptors would yield a perceptually meaningful multidimensional shape space due to their complementary nature. (**C**) Some shape descriptors are highly sensitive to rotation (e.g., Major Axis Orientation), while (**D**) other descriptors are highly sensitive to bloating (e.g., Solidity). (**E**) Over 100 shape descriptors were evaluated in terms of how much they change when shapes are transformed ('sensitivity').

We illustrate this for two transformations: rotation and bloating (**Fig 1C and 1D**). Specifically, we transformed one exemplar from each of 20 different animal categories (e.g., birds, cows, horses, tortoise) with bloating and rotation transformations of varying magnitudes (see **Methods**: *Sensitivity/robustness analysis to transformation*). We find that the different descriptors are differentially sensitive to the transformations. Some shape descriptors (e.g., *solidity* which measures the proportion of the convex hull that is filled by the shape; [45]; **Fig 1C and 1D**) are entirely invariant across rotations, while others (e.g., *major axis orientation*) are sensitive to object orientation. Yet descriptors invariant to rotation may be highly sensitive to other transformations, like bloating (**Fig 1C–1E**). Similarly, adding noise to a shape's contour strongly affects curvature-based metrics, while only weakly affecting the shape's major axis orientation (**S1 Fig**). In **Fig 1E**, we plot how sensitive 109 different shape descriptors are to the changes introduced by rotation and bloating, highlighting the descriptors identified in **Fig 1B**. Interestingly, for these transformations, there is a trade-off in sensitivity such that descriptors that are highly sensitive to bloating (e.g., *solidity*) tend to be less sensitive to rotation, and vice versa (e.g., *major axis orientation*). In other words, as expected, different shape features have complementary strengths and weaknesses. More generally, the plot shows the wide range of sensitivities across different shape metrics, indicating that depending on the context or goal, different shape features may be more or less appropriate [36, 46]. Note, of course, that were we

to choose other transformations (e.g., **S1 Fig**), the pattern would be different: here we selected rotation and bloating simply for illustrative purposes.

The key idea motivating our model is that human vision may resolve the conflicting demands of robustness and sensitivity by representing shape in a multidimensional space defined by many shape descriptors (**Fig 1B**). While it is widely appreciated that visual shape representations are likely multidimensional, in practice computational implementations of shape similarity metrics have typically used only a small number of quantities to capture relationships between shape [46–48]. As opposed to previous work, here we provide a data-driven implementation that determines the dimensions needed to capture variance among natural animal shapes. The approach does in fact contain many more dimensions than proposed previously, sufficiently accounts for human shape similarity, and provides a novel baseline metric against which more sophisticated computations can be compared.

We do not intend the model to be a simulation of brain processes, but as an efficient means to predict visual shape similarity judgments. It is unlikely the brain computes the specific model features considered here, most of which are taken from previous literature (see Supplemental **S1 Table**). Indeed, there are infinitely many other shape descriptors that could also be considered. Rather, we see the model as a concrete implementation of the idea that human shape similarity can be predicted by representing shape using multiple, complementary geometrical properties. Indeed, once many features are considered, the specific details of any given feature become progressively less important (although we do not imply that all shape descriptors are equally useful for any given task).

## Results and discussion

### Analysis of real-world shapes

Different shape descriptors are measured in different units, so to combine the features into a consistent multidimensional space requires identifying a common scale. Given the importance of real-world stimuli for human behavior, we reasoned that the relative scaling of the many feature dimensions likely reflects the distribution of feature values across real-world shapes. We therefore assembled a database of over 25,000 animal silhouettes and for each of them measured >100 shape descriptors (**Methods:** *Real-world shape analysis*). For every pair of shapes, we computed the distances between each descriptor (scaled by their largest distance across the whole animal dataset; **Fig 1B**) and then combined the features into a single metric, yielding a multidimensional space. This space exhibited a prominent shape-based organization with nearby locations sharing similar shape characteristics. For example, approximately elliptical animals like rabbits, fish, and turtles lie near together (bottom left of **Fig 2A**), while spindly thin-legged shapes (e.g., spiders; see insets in **Fig 2A**) are found in the opposite corner of the space.

As an initial indicator of how well the features account for perceptual similarity with familiar objects, we took a subset of animal shapes, and measured human similarity judgements (**Fig 2B and 2C**) using a multi-arrangement method [49]. We find that the mean perceived similarity relationships between shapes were quite well predicted by distance in this feature space (**Fig 2D–2F**, $r = 0.63$, $p < 0.01$) suggesting that the 109 shape descriptors explain a substantial portion of the variance in human shape similarity of familiar objects. We suggest that at least some of the remaining variance is likely to be due to using familiar objects, for which high-level semantic interpretations are known to influence similarity judgments [50–54]—here, the perceived classes to which the animals belong, rather than their pure geometrical attributes.

We also find that many of the shape descriptors correlate with one another, yielding 22 clusters of related features (using affinity propagation clustering; [55]). Using Multidimensional Scaling across the 25,712 animal shape samples, we find that 22 dimensions account for more

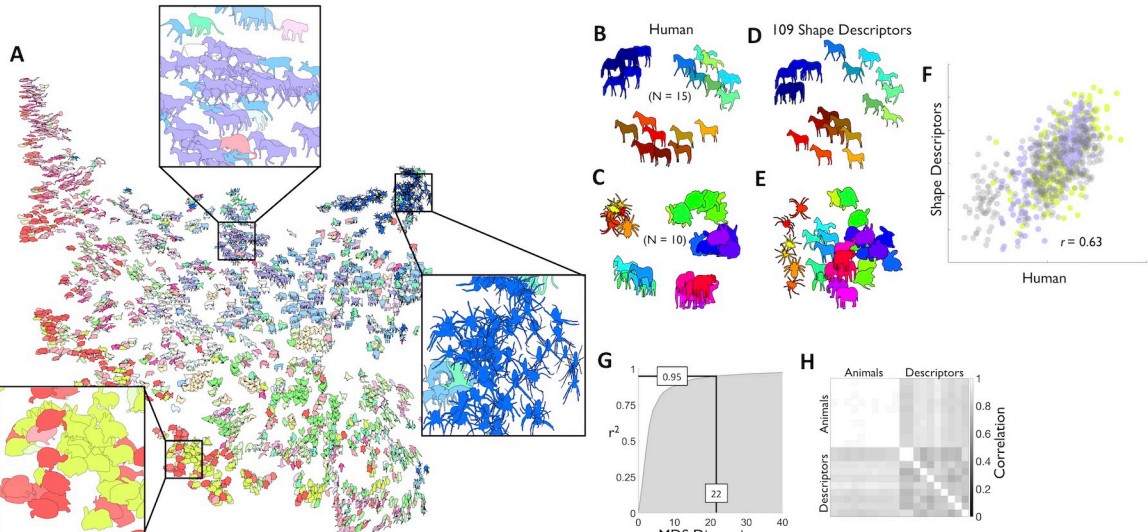

**Fig 2. The high-dimensionality of real-world shapes.** (**A**) t-SNE visualization of 2000 animal silhouettes arranged by their similarities according to a combination of 109 shape descriptors. Colour indicates basic level category. Insets highlight local structure: bloated shapes with tiny limbs (left); legged rectangular shapes (middle); small spiky shapes (right). To test whether human shape similarity is predicted in the high-dimensional animal space, we gathered human shape similarity judgments on horses (purple), rabbits (yellow), and other animals. (**B**) Human similarity arrangements of horse silhouettes, and (**C**) of silhouettes across multiple categories of animals (multidimensional scaling; dissimilarity: distances, criterion: metric stress). Similarity arrangement for (**D**) horse silhouettes and (**E**) multiple categories of animals in the full model based on 109 shape descriptors (multidimensional scaling; dissimilarity: distances, criterion: metric stress). Shapes with same colour across B and D or C and E are also the same. (**F**). Human arrangements correlate with the model for horse (purple), rabbit (yellow), and multiple animal silhouettes (gray) ($r = 0.63$, $p < 0.01$). (**G**). Across 25,712 animal shapes, 22 dimensions account for >95% of the variance (multidimensional scaling; dissimilarity: distances, criterion: metric stress). We call these 22 dimensions ShapeComp. (**H**) The space spanned by these ShapeComp dimensions regularly occurs across combinations of different animal sets ('Animals') and shape descriptors ('Descriptors'). The pairwise distances across 200 test shapes are highly correlated across ShapeComp computed from 10 different sets of 500 randomly chosen animal shapes ('Animals'), and also, but to a lesser degree, across 10 different sets of randomly selected shape descriptors ('Descriptors'; 55 out of 109).

than 95.05% of the variance (**Fig 2G**), whereas the first dimension accounts for only 18.54% of the variance. We refer to this reduced 22-D space as ShapeComp (**Fig 1B**), and it is this model that forms the basis of the majority of our subsequent analyses.

ShapeComp's dimensions are composites (i.e., weighted linear combination) of the original shape descriptors, which makes the model fully interpretable, unlike other model classes (e.g., neural networks, whose inner functioning researchers still struggle to interpret [56, 57]). Although we do not believe the brain explicitly computes these specific dimensions, they do organize novel shapes systematically (see **Results:** *Using Generative Adversarial Networks to create novel naturalistic outlines*). However, because MDS creates a rotation invariant space, individual dimensions should not be thought of as 'cardinal axes' of perceptual shape space. Rather it is the space as a whole that describes systematic relationships between shapes. Thus, while thinness and leggedness may not be coded in ShapeComp as unique or cardinal dimension, as hinted in **Fig 2A**, thin shapes thinner are nearer to other thin shapes than to thick shapes, and shapes with legs (e.g., spiders) tend to be nearer to other legged shapes (e.g., centipedes) than to those with no legs (e.g., fish). It is important to note that our focus on relative similarities between items—rather than putative 'cardinal dimensions' of perceptual space—is not specific to ShapeComp, but is rather a core assumption of many studies and analyses that compare measurements of human perception with models or brain activity [58–69]. Indeed, while it may be possible to define 'cardinal perceptual dimensions' for limited synthetic stimulus arrays [47, 48, 70, 71], we would question whether there are any meaningful axes that span the complete range of complex naturalistic shapes.

Given these 22-dimensions are composites of the original 109 features, one might ask what are that best original features? **S2A–S2H Fig** shows that several of the original features are highly correlated to each of first 8 dimensions of ShapeComp (which already accounts for greater than 85% of the variance in animal shapes), suggesting that many features tap into complementary aspects of shape. Thus, ShapeComp will not undergo major changes if one of the original features is removed. Similarly, **S3A–S3H Fig** shows several poor predictors that presumably vary less across the animal silhouette database than other features. **S2I** and **S3I Figs** show the best and worst features across the full 22D space, respectively. The Shape Context and summaries based on the Shape Context (e.g., histogram of chord lengths) were most predictive of ShapeComp, while the skeletal and low frequency Fourier descriptors were least predictive. (Note, however, the less predictive shape descriptors are likely still useful for shape similarity. Firstly, the features posited here are partial summaries of the original shape descriptors. For example, one feature taken from the shape skeleton was the number of ribs. There are likely a number of other ways to summarize the shape skeleton that may be more sensitive to change in animal shapes across our database. Secondly, it is likely that such features play an important role in finer shape discrimination judgments that go beyond ShapeComp's 22-dimensions.)

One caveat that concerns the usefulness of any high-dimensional space is its reproducibility: Does the ShapeComp space come together by chance, e.g., based on a specific animal dataset, or does ShapeComp capture regularities that tend to occur across animal shapes more generally? We find that ShapeComp's space is not brittle, but robust across the selection of animal shapes or shape descriptors (**Fig 2H**). Specifically, the distance relationship across 200 test shapes is highly related when ShapeComp is computed in (1) different random subsets of animal shapes ($0.98 \leq r \leq 0.99$; relationship across 10 different sets), and also, but to a lesser degree, in (2) different random combinations of shape descriptors ($0.69 \leq r \leq 0.93$; relationship across 10 different sets). In addition, despite removing the most predictive features of ShapeComp (i.e., 11 features related to the Shape Context and its summaries; listed as descriptors 29–31, and 52–59 in **S1 Table**) pairwise, distances between shapes remain highly correlated ($r = 0.77$, $p<0.01$). Thus, ShapeComp appears to capture a high-dimensional understanding of shape that tends to be somewhat independent across the specific selection of animal shapes or even shape descriptors.

## Using Generative Adversarial Networks to create novel naturalistic outlines

To reduce the impact of semantics on shape similarity judgments, we next created novel (unfamiliar) shapes using a Generative Adversarial Network (GAN) trained on the animal silhouette database (see **Methods**: *GAN Shapes*). GANs are unsupervised machine learning systems that pit two neural networks against each other (**Fig 3A**), yielding complex, naturalistic, yet largely unfamiliar novel shapes. The GAN also allows parametric shape variations and interpolations in a continuous 'shape space' (**Fig 3B–3D**). We tested whether GAN shapes evoked percepts of specific familiar objects by comparing human categorization responses of 100 randomly selected GAN shapes versus 20 animal shapes. As desired, the most incompatible responses across observers were found for GAN shapes (**Fig 3E**), allowing us to identify stimuli with weak semantic associations, and thus reduce the impact of semantics on shape similarity judgments. Overall, the GAN shapes appear 'object-like', but observers agree less about their semantic interpretation, compared with animal shapes, making them better stimuli for assessing pure shape similarity.

With the GAN's generator network we can synthesize arbitrary numbers of novel naturalistic looking shapes, and estimate their coordinates in ShapeComp. This serves both to visualize

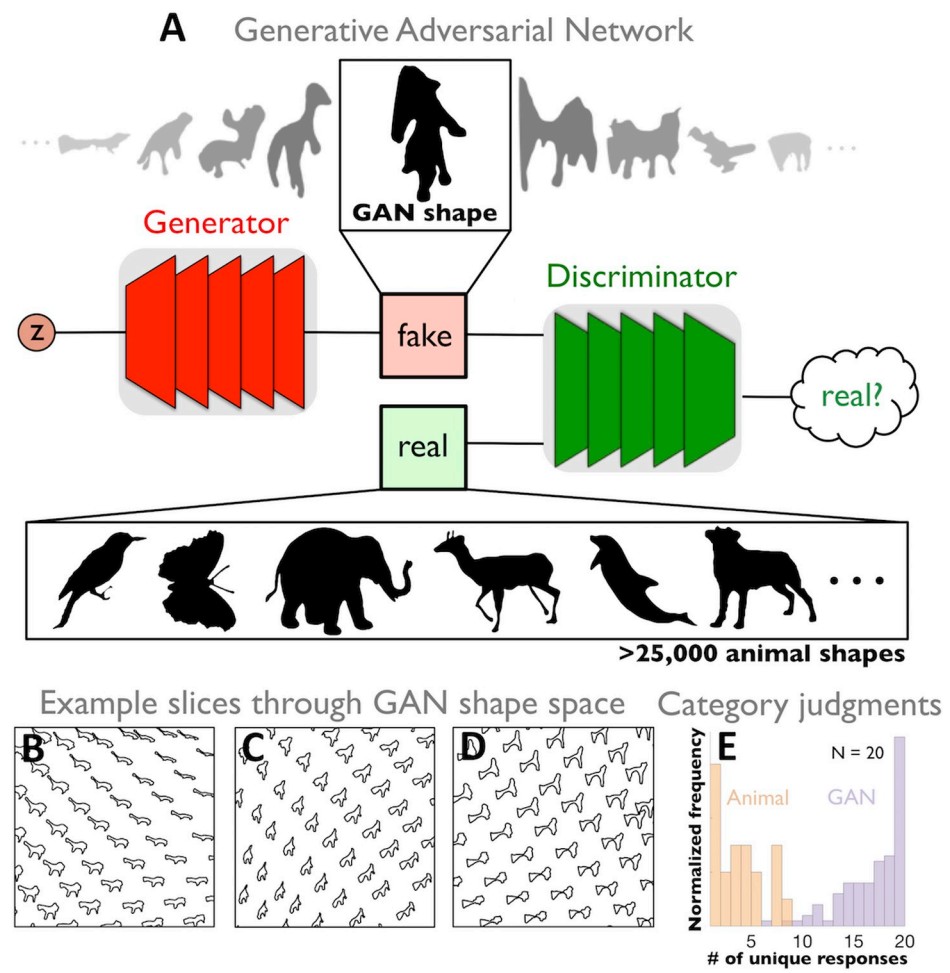

**Fig 3. GANs produce novel naturalistic shapes.** (**A**) Cartoon depiction of a Generative Adversarial Networks (GANs) that synthesizes novel shape silhouettes. GANs are unsupervised machine learning systems with two competing neural networks. The generator network synthesizes shapes, while the discriminator network, distinguishes shapes produced by generator from a database of over 25,000 animal silhouettes. With training, the generator learns to map a high-dimensional latent vector '$z$' to the natural animal shapes, producing novel shapes that the discrimantor thinks are real rather than synthesized. Systematically moving along the high-dimensional latent vector $z$ produces novel shape variation and interpolations across a shape space (**B, C,** and **D**). (**E**) A normalized histogram with the number of unique responses across 100 GAN shapes and 20 animal shapes shows that category responses across GAN shapes tend to be much more inconsistent across participants than animal shapes, confirming that GAN shapes appear more unfamiliar than animal shapes.

the dimensions of ShapeComp, and test their role in perceptual shape similarity. As discussed above, we emphasize the importance of considering ShapeComp as a composite multidimensional space and caution against attempts to interpret individual dimensions as 'cardinal axes' of shape space. Nevertheless, to understand the space better, it is still helpful to visualize the shape characteristics described by individual dimensions. **Fig 4** shows such a visualization. GAN shapes vary in the first 6 (out of 22) MDS dimensions while the remaining dimensions are held almost constant. At least the first few dimensions are systematically organized with distinctive and different types of shape at opposite ends of each scale. However, much like the properties of receptive fields in mid- and high-level visual areas, it is not always easy to verbalize the properties underlying each MDS dimension. For example, dimensions 1 and 3 appear to modulate horizontal and vertical aspect ratio, respectively, but other factors like number

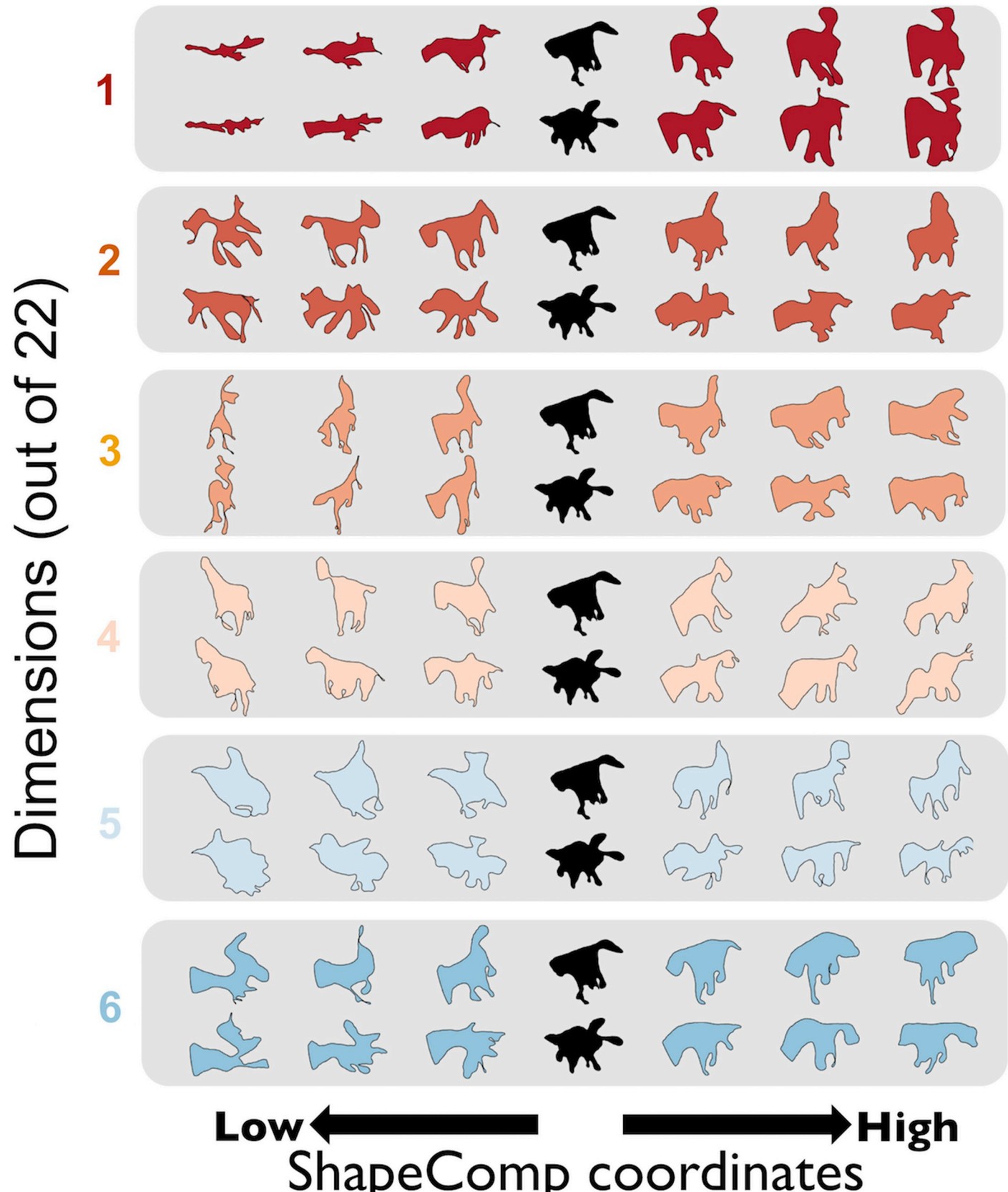

**Fig 4. Interpreting ShapeComp dimensions.** Example GAN shapes that vary along the first 6 MDS dimensions. Two shapes (in black) are varied along one dimension (in different colours, dimensions 1–6) while the remaining dimensions are held roughly constant. The different GAN shapes that varied in their MDS coordinates were optimized with a genetic algorithm from MATLAB's global optimization toolbox to reduce RMS error between a GAN shapes 22-D representation and a desired 22-D representation.

and extent of limbs also vary. Other dimensions appear to morph between specific types of shape or specific shape poses (e.g., a shape 'facing' left vs. right).

Having confirmed that GAN shapes had less clear semantics than the animal shapes, we next examined how well the model captures human perception of unfamiliar objects. Specifically, in the following sections, we sought to test more rigorously (a) whether distance in ShapeComp space predicts human shape similarity, (b) whether ShapeComp provides information above and beyond simpler metrics like pixel similarity, (c) whether human shape similarity relies on more than one ShapeComp dimension, and (d) whether ShapeComp identifies perceptual nonlinearities in shape sets.

## Distances in ShapeComp model predict human shape similarities for novel objects

A key criterion for any perceptual shape metric is that pairs of shapes that are close in the space (**Fig 5A, top**) should appear more similar than pairs that are distant from each other (**Fig 5A, bottom**). To test this, we generated 250 pairs of novel GAN shapes, ranging in their ShapeComp distance (i.e., predicted similarity), and asked 14 participants to rate how perceptually similar each shape pair appeared (**Fig 5B**). We find that distance in ShapeComp correlates strongly with the mean dissimilarity ratings across observers ($r = 0.91$, $p < 0.01$) showing that ShapeComp predicts human shape similarity very well for novel unfamiliar 2D shapes.

Still unclear, however, is whether ShapeComp captures aspects of human shape similarity perception better than standard benchmark metrics. There are some grounds for expecting that it might do. Because ShapeComp combines 109 different descriptors—which between them capture many distinct aspects of shape—it is likely that the model describes shape in a richer, more human-like way than conventional raw pixel similarity. Moreover, we can test whether ShapeComp is better at predicting shape similarity than any of its individual metrics. One challenge in comparing existing metrics and their role in human vision, is that the features tend to be strongly correlated with one another. The orthogonal (i.e., decorrelated) dimensions of ShapeComp allow us to confirm whether human shape similarity relies on linearly independent components of the original 109 shape descriptors.

## ShapeComp predicts shape similarity better than widely-used pixel similarity metrics

A standard way to measure the physical similarity between shapes is the Intersection-over-Union quotient (IoU; [37, 72]; **Fig 5C**). The method is one of the most widely used in computer vision and machine learning research as a benchmark to evaluate performance in segmentation [73–76], object detection [76, 77], and tracking [78, 79]. For similar shapes, the area of intersection is a significant proportion of the union, yielding IoU values approaching 1. In contrast, when shapes differ substantially, the union is much larger than the overlap, so IoU approaches 0. Despite its simplicity, similar pixel-based metrics have also been used extensively in perceptual and neuroscientific studies as a benchmark for physical similarity between objects or shapes [23, 52, 80–86].

To test whether human shape similarity can be approximated by such a simple pixel similarity metric or rather relies on more sophisticated mid-level features like those in ShapeComp,

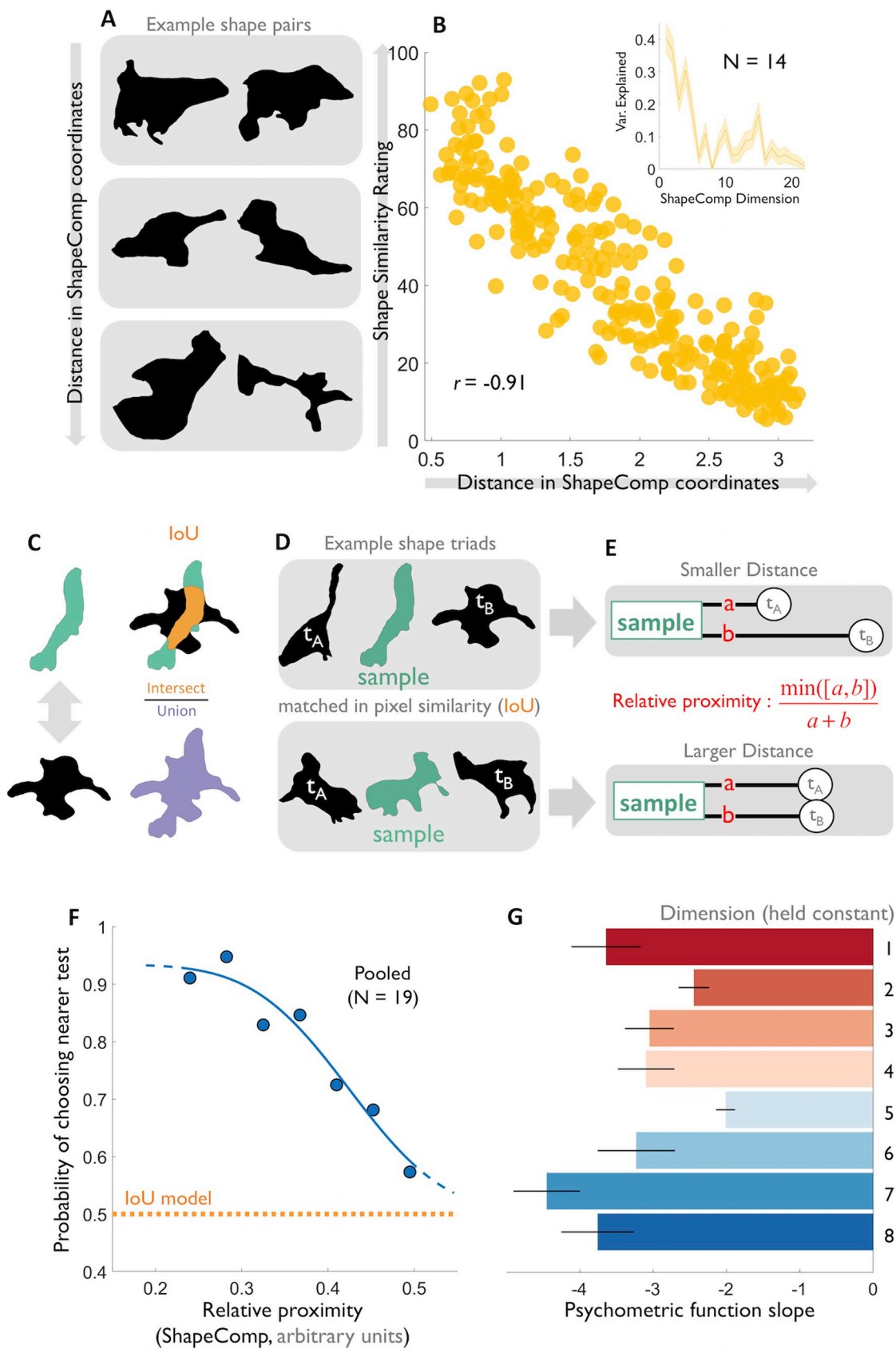

**Fig 5. ShapeComp predicts human shape similarity across small sets of shapes. (A)** Example shape pairs that varied as a function of ShapeComp distance. **(B)** Shape similarity ratings averaged across 14 observers for 250 shape pairs highly correlate with distance in ShapeComp's 22-dimensional space. Inset: The variance in the similarity ratings accounted for by the different ShapeComp dimensions. Many ShapeComp dimensions on their own account for some of the variance in human shape similarity ratings. Shaded error bars are estimated via 1000 bootstrapping across participant responses. **(C)** Pixel similarity was defined as the standard Intersection-over-Union (IoU; [37, 72]) **(D)** Observers viewed shape triads and judged which test appeared more similar to the sample. **(E)** ShapeComp distance between test and sample were parametrically varied but pixel similarity was held constant. **(F)** Mean probability across participants, that the closer of two test stimuli was perceived as more similar to the sample, as a function of the relative proximity of the closer test shape. Blue: psychometric function fit; orange: prediction of IoU model. **(G)** Results of experiment in which distances from test to sample were equated for one ShapeComp dimension at a time. Mean psychometric functions slopes were much steeper than predicted if observers relied only on the respective dimension. These results, and that the variance in the similarity ratings is accounted for by many ShapeComp dimensions, inset in **B**, support the idea that human shape perception is based on a high-dimensional feature space.

we created stimulus triplets, consisting of a *sample* shape, plus two *test* shapes, which were equally different from the sample shape in terms of IoU but which differed in ShapeComp distances (**Fig 5D** and **Methods:** *pixel similarity triplets*). This allowed us to isolate the extent to which ShapeComp predicted additional components of shape similarity, above and beyond pixel similarity. The magnitude of the difference between tests and sample in ShapeComp was varied parametrically across triplets, so that sometimes one test was much nearer to the sample than another test (**Fig 5E**). Nineteen new participants viewed the triplets and were asked which of the two test shapes most resembled the sample on each trial. If shape perception is perfectly captured by IoU, the two test stimuli should appear equally similar to the standard, yielding random responses (**Fig 5F** orange line). However, we find that the slope of a psychometric function fitted to the observers' judgments is significantly steeper than zero (**Fig 5F** blue line; $t = -7.63$, df = 18, $p < 0.01$). This indicates that ShapeComp correctly predicts which of the two shapes was more similar to the standard even when pixel similarity is held constant. Consistent with previous works [23, 44, 80–85, 87], this confirms that human shape similarity relies on more sophisticated features than pixel similarity alone. Thus, ShapeComp provides a concrete implementation of the widely held belief that such metrics are insufficient, despite their continued widespread use in the literature.

## ShapeComp captures multidimensional nature of human shape similarity

Although a standard model of comparison in human perception, pixel similarity is a rather simple model. Many better alternative models are encompassed in the many dimensions of ShapeComp, where each dimension shows shape variation along an orthogonal dimension. To verify that human shape similarity considers multiple aspects of ShapeComp (i.e., relies on more than a single of ShapeComp's orthogonal dimensions), we generated triplets in which the test shapes were equated to a given sample shape in terms of one of ShapeComp's 22 dimensions but varied in terms of the remaining dimensions. The same nineteen participants as in the pixel similarity experiment were shown these triplets and again reported which test shape appeared most similar to the sample. If shape perception is entirely captured by any single dimension, the two test stimuli should appear equally similar to the sample, yielding random responses. Yet **Fig 5G** shows that fitted psychometric function slopes were significantly steeper than zero. This confirms that human shape perception relies on more than a single ShapeComp dimension—when each dimension was held constant, the variations in the remaining dimensions dominated perception.

We also re-analyzed the ratings from **Fig 5B**, comparing the human judgments to each ShapeComp dimension. Each dimension on its own accounted for only a small portion of the variance (inset in **Fig 5B**), again indicating that human observers rely on more than one

ShapeComp dimension. Together, these results confirm that ShapeComp successfully captures the inherently multidimensional representation of shape in human vision.

## Identifying perceptual nonlinearities in shape spaces of novel objects

So far, our evaluations of ShapeComp have focused on judgments of relative similarity among small sets of stimuli (e.g., of the form "is shape A more similar than shape B is to shape C"). Yet, an important test for any human shape similarity metric is its ability to predict richer similarity relationships within arrays of multiple shapes. To assess this, we tested how well Shape-Comp identified perceptual non-uniformities in shape spaces generated with the animal-trained GAN.

The top row in **Fig 6** shows four example 2D GAN shape arrays sampled uniformly across 3 radial distances (**Fig 6A and 6B**) or along a triangular grid (**Fig 6C and 6D**). The second row in **Fig 6** shows that ShapeComp's predicted arrangement of these shapes (in 2D) is non-uniform with a substantial compression around certain items (e.g., the thinner shapes in Shape Set **A**). Using a multi-arrangement task (**Methods**), we find that human perceived similarities within these arrays were similar in terms of the relative ordering of shapes and, in many shape sets, also showed the nonuniformities predicted by ShapeComp (e.g., compression of thinner shapes in Shape Set shown in **Fig 6A**; mean responses from 16 participants: third row in **Fig 6**).

To test how well the model predicts participants' responses, it is instructive to consider the extent that the perceptual distortions (i.e., deviations from the uniform GAN space) predicted by ShapeComp predict human shape similarity better than would occur by chance (i.e., under a random model). To do so, we defined and measured distortions between shape arrays as differences between two similarity matrices—each standardized to have unit variance—where larger differences lead to larger distortions. To test whether ShapeComp is better than a random model, we developed a GAN+noise model that distorts the original GAN space by adding random Gaussian perturbations to the original GAN latent vector coordinates. We set the noise level of the model to maximize its chance of accounting for the human distortions by matching the overall distance of the noise perturbations from the original GAN space with the overall perturbations of the human observers (from the original GAN space). Across four shape sets where GAN and ShapeComp spaces tended to be less correlated with one another ($0.59 < r < 0.75$), perceptual distortions in GAN space by individual observers were better accounted for by ShapeComp than the GAN+noise model (**Fig 6E**). Further, shape sets with more diversity across their shapes (i.e., that varied more in terms of their underlying Shape-Comp coordinates) were better predictive of how well ShapeComp distortions matched humans: Greater variance in ShapeComp across a shape set lead to more overlap with humans ($r = 0.72$, $p < 0.01$; **Fig 6F**). Thus, ShapeComp correctly predicted the direction of perceptual nonlinearities in the GAN space. This is striking given that the GAN arrays and ShapeComp are highly correlated, and thus already share much of the variation across their arrangements of the shape sets.

## Deriving perceptually uniform shape spaces of novel objects

To examine shape perception independently of high-level vision, previous work controlled for perceptual shape similarity through time-consuming measurements (e.g., [52, 54, 86, 88–90]). With the ability to measure perceptual non-uniformities in hand—and as a second test of the ability to predict human shape similarity perception in multi-shape arrays—we evaluated ShapeComp's suitability for automatically creating perceptually uniform arrays of novel objects. To do this, we searched for uniform arrays in the GAN's latent vector representation that were

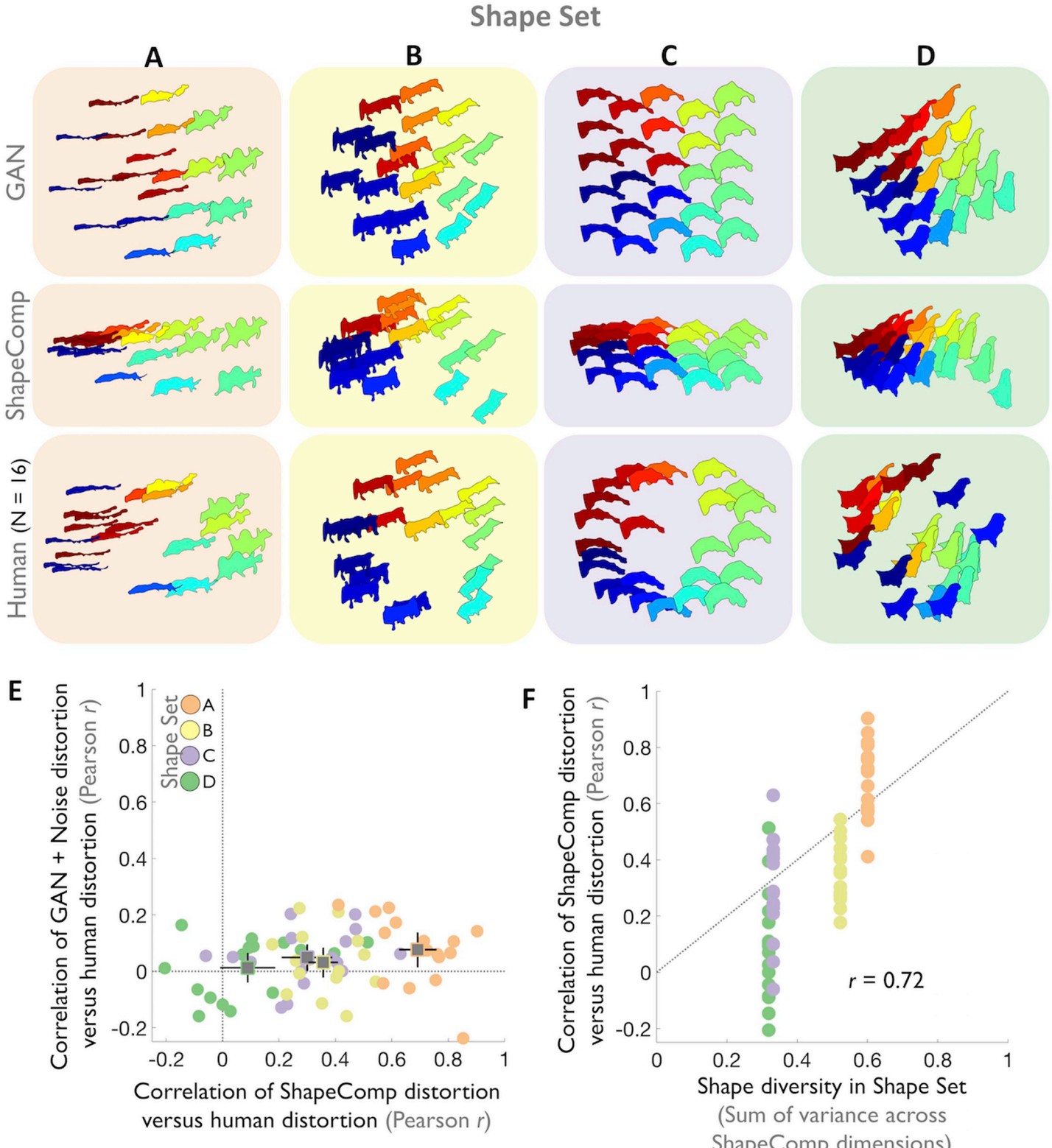

**Fig 6. ShapeComp predicts perceptual distortions in human shape similarity across shape arrays.** Four example shape sets (**A, B, C, D**) sampled uniformly in GAN space (top row). To test whether subtle perceptual distortions in humans were systemically deviated away from GAN space towards ShapeComp, these shape sets were selected such that the pairwise distances of shapes in ShapeComp varied slightly from GAN (with Pearson correlation values between $0.5 < r < 0.75$). The arrays are distorted by ShapeComp (second row) in similar ways to humans (third row; mean across 16 participants). Across arrangements, shapes with same colour are also the same.

(**E**) Non-uniformities for individual participants (dots) in 4 shape sets (**A-D**, colours). Squares show average across subjects for given set, where error bars show ± 2 standard errors. ShapeComp accounted for perceptual distortions away from the original GAN coordinates better than GAN+noise model. (**F**) Correlation of ShapeComp distortion with human distortion as a function of the diversity of shapes across the shape set (measured as cumulated variance in shape set across ShapeComp dimensions). Human distortions better line up with ShapeComp when there is more diversity across shape sets as predicted by ShapeComp. Grey reference line shows y = x.

highly correlated with ShapeComp ($r>0.9$), and had participants arrange these sets based on their similarity. The top row in **Fig 7** shows four arrays (**Fig 7A–7D**) that ShapeComp predicts should be arranged almost uniformly. Human similarity arrangements (mean response from 16 participants; second row in **Fig 7**) are mostly consistent with ShapeComp in terms of the relative ordering of the shapes. Across three of the four different shapes sets, human responses are nearly indistinguishable from the predictions of ShapeComp, given the inherent noise across observers (**Fig 7E**). In the one case that the model deviates significantly from humans (Shape Set in **Fig 7B**), humans tend to weigh certain features (e.g., the apparent tail of the shape) more heavily than ShapeComp. One way the model may improve its prediction is by using a different (e.g., fitted) weighted combination of the 22 ShapeComp dimensions. Despite this one deviation, these results show that combining the high-dimensional outputs of the GAN with ShapeComp is a useful tool for automatically creating a large number of perceptually uniform shape spaces.

## ShapeComp network

Given the usefulness of creating shape arrays for carefully controlled stimulus sets, and for neuroscientific investigations on shape, we make available several tools (**Fig 8**) that allow experimenters to (1) compute a given shape's ShapeComp coordinates, and (2) create many novel shape sets using the GAN. The method can be used to create novel shape arrays with controlled shape similarity relationships (**Fig 9**), or can be applied on existing shapes to quantify their shape similarity (e.g., **Fig 10**).

Although the features underlying ShapeComp are both image computable and interpretable, in practice, the codebase is convoluted as it draws on many different sources. Moreover,

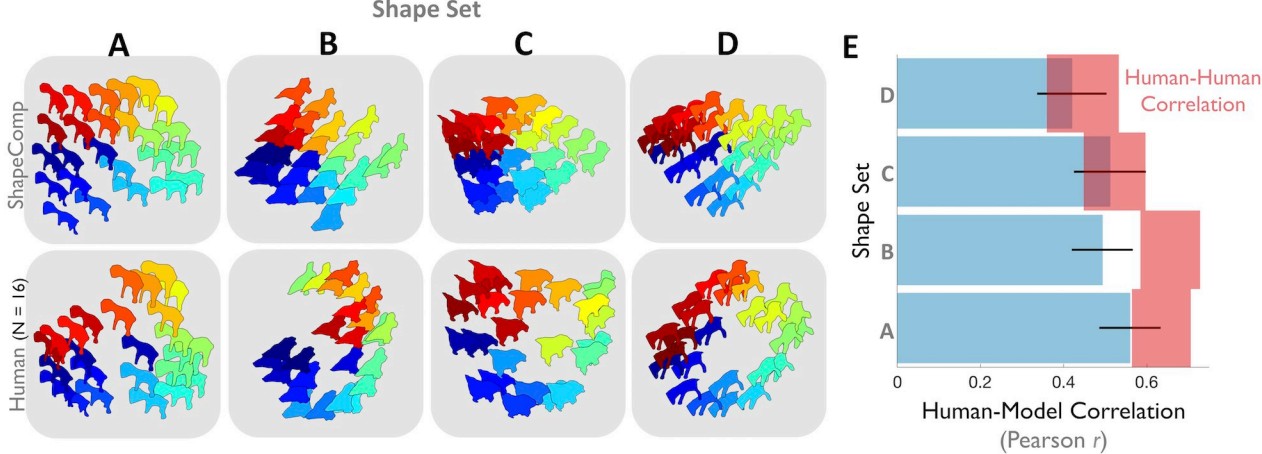

**Fig 7. ShapeComp predicts perceptual uniformities in human shape similarity across shape arrays. (A,B,C,D)** The top row shows four example 2D shape arrays that are roughly uniform in ShapeComp and highly correlated to the GAN arrangement (r>0.9). The bottom row shows the mean arrangement by 16 human observers. (**E**) In 3 out of 4 shape sets that are highly correlated in terms of GAN and ShapeComp arrangements, human responses are nearly indistinguishable from the predictions of ShapeComp (blue), given the inherent noise across observers measured as the lower noise ceiling (red; 95% confidence interval showing correlation of each participant's data with mean of others). Error bars (in black) show 95% confidence interval around human-model correlation.

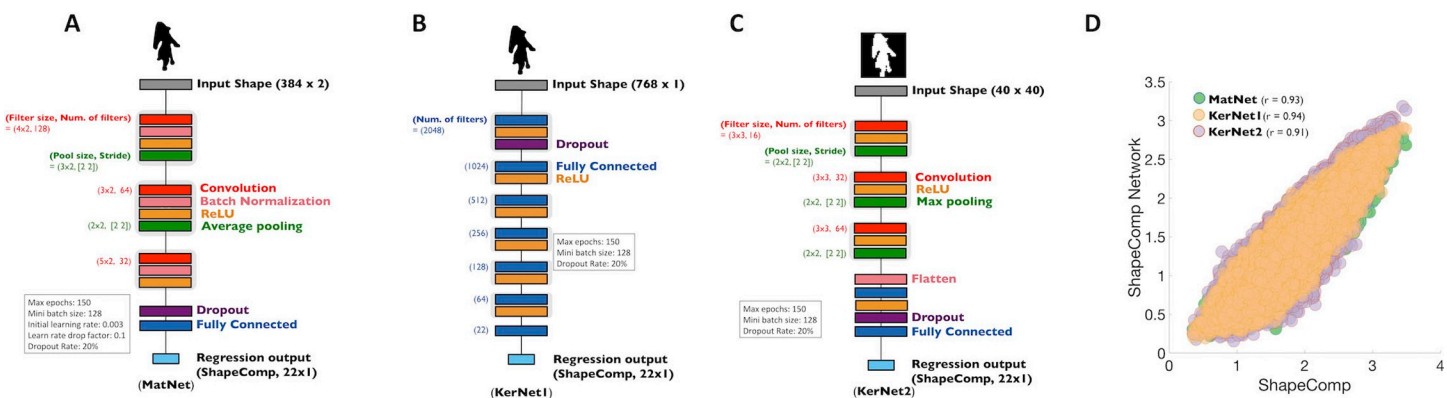

**Fig 8. ShapeComp neural network for estimating a shape's 22-Dimensional ShapeComp coordinates.** Neural networks in **(A)** MATLAB (MatNet) and **(B)** Python (KerNet1) were trained on 800,000 shapes to get as input the shape *x,y* coordinates and output the 22D high-dimensional shape space. **(C)** Kernet2, also in Python, was trained to output the ShapeComp coordinates from 40×40 image patches. **(D)** The networks 22-dimensional distances across all pairwise comparisons of 1000 untrained shapes are highly correlated to the pattern of distances from the original ShapeComp solution.

the computation of all 109 features along with pairwise comparisons with values pre-computed from a large dataset of stored animal shapes is too slow for real-time applications. Furthermore, as argued above, individual features are less important than the space spanned by them in concert. Thus, to consolidate ShapeComp into a single, high-speed model, we trained a multi-layer convolutional neural network on 800,000 GAN shapes that spanned the high-dimensional space. We trained three versions (**MatNet**, **KerNet1**, **KerNet2** in **Fig 8A–8C**) to provide cross platform capabilities. **MatNet** and **KerNet1** are networks trained in MATLAB and Keras, respectively, that use the shape's x,y coordinates as input. **KerNet2**, also trained in Keras, uses a 40x40 binary image of the shape as input. Each network takes shapes as input and

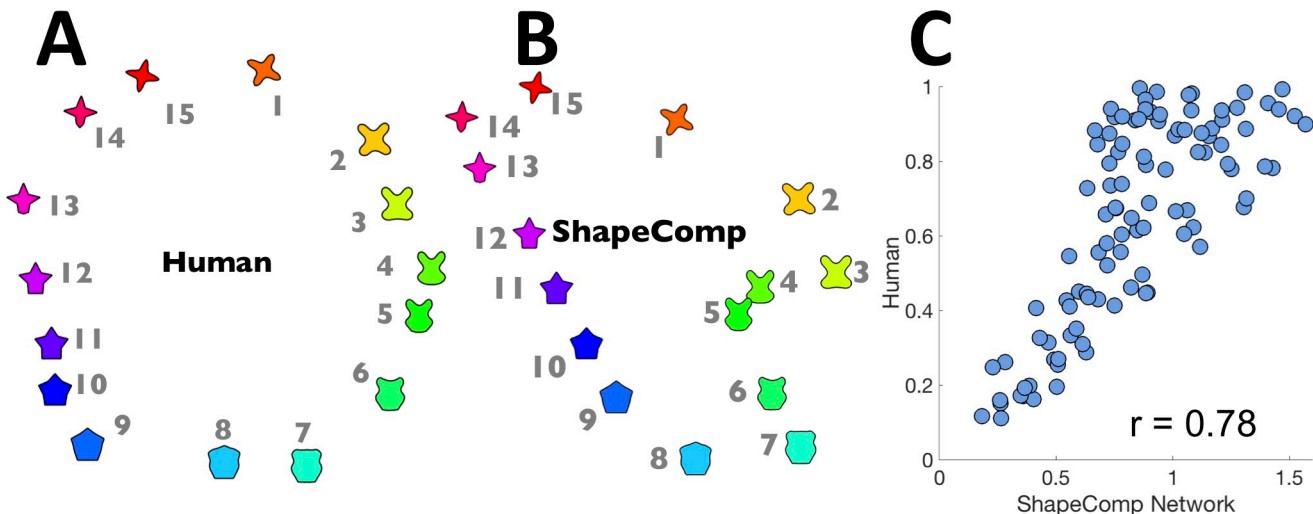

**Fig 9. Using ShapeComp to evaluate shape similarity in existing shape sets.** Even with novel shapes from, as an example, the **(A)** validated circular shape space set (human data; from [90]), **(B)** ShapeComp's predictions show many similarities to humans. While ShapeComp's arrangement is more compressed, ShapeComp correctly predicts (i) large gaps between shapes 1 and 15, and 1 and 2, (ii) the circular nature of the data set, (iii) subjective difference between 1 and 11 is smaller than between 14 and 8, yielding the elongated arrangement. **(C)** Correlation between ShapeComp and human similarity judgments for the distances between all possible (105 pairs) ($r = 0.78$, $p < 0.01$). Given the noise uncertainty across observers–which is unknown for the circular shape set—ShapeComp appears to be a good model of human behaviour. Note, given that some shapes in the circular shape set (e.g., 5 or 6) have multiple minimum *x*-values, we used KerNet2 which is based on images to compute the ShapeComp solution.

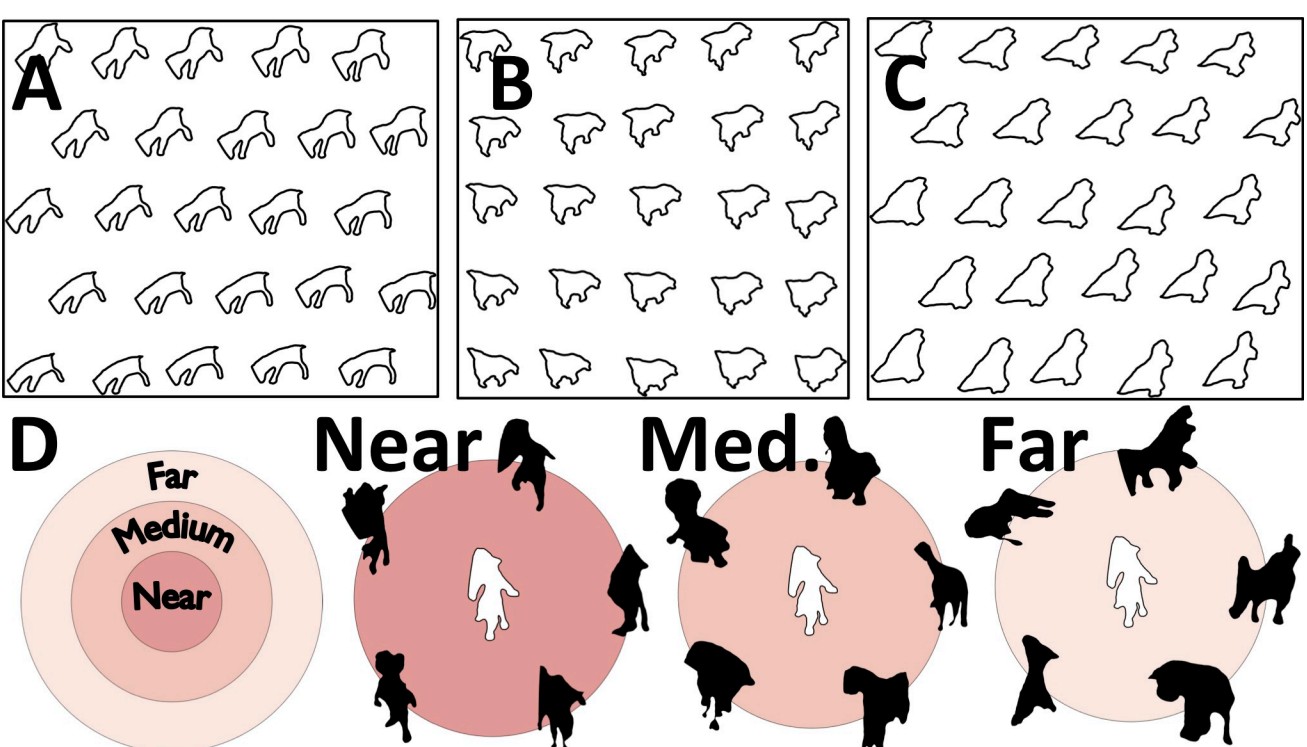

**Fig 10. Synthesizing perceptual uniform shape spaces.** ShapeComp paired with GAN can be used to create perceptually uniform shape spaces (**A-C**) along a triangular (**A, C**) or uniform (**B**) grid or in selecting test shapes that have similar shape similarities (**D**, near, medium, or far in terms of their distances in ShapeComp) to the central sample shape.

outputs a 22-dimensional vector, representing the values of each of the dimensions of Shape-Comp (see also **Methods:** *Shape to ShapeComp Network*).

The average error of the network in estimating ShapeComp coordinates (in untrained shapes) is within the range of ShapeComp values that human observers tend to judge as very similar. Specifically, the network produced a mean error in ShapeComp's units of 0.45 across 150,000 untrained shapes. For comparison, humans rate shapes within 0.5 ShapeComp units as highly similar (see **Fig 5B**), indicating that the neural network provides sufficiently good approximation to ShapeComp for most practical purposes. More important than absolute deviation between ShapeComp coordinates is how ShapeComp captures the relationship between shapes. We find that the network's predicted distances across the upper triangular matrix of all pairwise combinations in 1000 untrained shapes is highly related to ShapeComp (**MatNet**; $r = 0.93$, $p < 0.01$; **KerNet1**; $r = 0.94$, $p < 0.01$; **KerNet2**; $r = 0.91$, $p < 0.01$), which is significantly larger than the correlation of human shape similarity judgements across different observers in much smaller shape arrays (**Fig 7E**).

The networks allow experimenters to identify where arbitrary shapes lie within the 22D ShapeComp space. For example, applied to artificial stimuli like the human-validated circular space shape set (from Li et al., 2019; reproduced in **Fig 9A**), the networks yield a Shape-Comp solution (in **Fig 9B**) that is highly related to human judgements (**Fig 9C**), thus making the network an efficient and quick way to measure similarity across arrays or pairs of shape. Paired with a shape generation tool (here, the GAN's generator network), the ShapeComp networks allow the automatic creation of many perceptually uniform shape spaces (**Fig 10**).

## ShapeComp predicts human shape similarity better than object recognition convolutional neural networks (CNNs) for novel shapes

Although shape is thought to be the most important cue to human object recognition, its role in artificial CNN object recognition is less clear. Some work observes that the networks are good models of human shape perception [91] while other studies note that conventional CNNs have some access to local shape information in the form of local edge relations, but they have no access to global object shapes [92, 93], and are typically biased towards textures [94]. Kubilius et al. [91] showed that GoogLeNet [95] is highly consistent with human object categorization based on shape silhouette alone, and showed how similarity in the outputs from its last layer clearly groups such silhouettes into object categories (e.g., man-made versus natural). It is therefore interesting to ask how well such object recognition neural networks predict human similarity judgments of novel objects like those we used for testing our participants and the ShapeComp model. We tested this by deriving predicted shape similarity from various pre-trained networks, for the novel GAN shapes from our rating experiment (in **Fig 5B**) and our similarity arrangements (in **Fig 7**). Following Kubilius et al. [91], we defined network shape similarity as Euclidean distance in their final fully-connected layer (with 1000 units). We find all the networks we considered were substantially less predictive of human shape similarity than ShapeComp, both in pairs of shapes and across sets of shapes (**Fig 11**). For example, Shape-Comp, was much better at predicting human shape similarity than GoogLeNet in pairs of novel shapes (**Fig 11A**) and across shape sets (**Fig 11B**), highlighting fundamental differences in the computation of shape by object recognition neural networks and humans. Even the best performing of the networks we tested (Resnet101) correlated poorly with human judgments compared to ShapeComp, despite its vastly larger feature space. Together these findings suggest that the ability to label objects in natural images is not sufficient to account fully for human shape

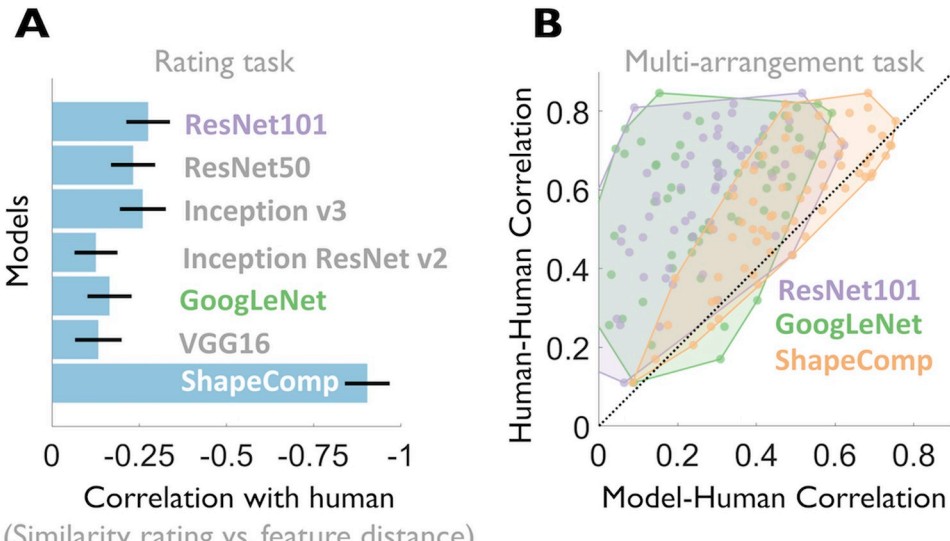

**Fig 11. Model comparison.** ShapeComp is more predictive of human shape similarity than standard object recognition neural networks across pairs of novel GAN shapes and shape sets. In (**A**) models are compared to human shape similarity ratings across pairs of shapes (data from **Fig 5B**). In (**B**) models are compared to individual observers' similarity arrangements (data from **Fig 7**). For any given shape set, each human observer's similarity matrix was correlated with the mean of the other observers (y-axis) and several models (ResNet101, GoogLeNet, or ShapeComp). The black line shows when an observer is equally correlated to other observers and the model. Only ShapeComp approaches this line, showing that it is a better model of human shape similarity across novel shape sets. Network shape similarity was defined as Euclidean distance in their final fully-connected layer (with 1000 units).

similarity judgments. We speculate that the nature of the shape computations in supervised object recognition neural networks trained on thousands of natural images is likely one of the many reasons why they fail to generalize like humans do, often incorrectly classifying cartoon depictions of images that even children with little experience easily classify. Consistent with this idea, increasing shape bias in these object recognition networks improves their accuracy and robustness [94].

## General discussion

Many previous studies have sought to measure shape similarity for both familiar and unfamiliar objects [23, 52–54, 82, 88, 90, 96–99]. Despite this, the representation of shape in the human visual system remains elusive, and the basis for shape similarity judgments remains unclear. In part, this is due to the numerous potential shape descriptors proposed in the past, including simple metrics, like solidity [36], and contour curvature [39], and more complex metrics like shape context [38], part-based ones [1, 85], Fourier descriptors [41, 100, 101], radial frequency components [82, 102], shape skeletons [40, 46, 98, 99, 103–107], linearity [108] convexity [109–112], triangularity [113], rectilinearity [114], information content [115, 116] and models based on generalized cylinders for describing 3D animal-like objects [117]. While it is widely believed that human shape representations are multidimensional, to date there has been no comprehensive attempt to implement this idea in a concrete image-computable model. Moreover, the continued widespread use of relatively simplistic pixel-based similarity measures [23, 52, 73–86] points to a significant unmet need for a standard alternative model. The main contribution of this study is to provide such a model.

Which features does the brain use to represent and compare shapes? It is important to emphasize that our goal was not to develop a process model of shape representation in the human brain, but rather to develop an image-computable model that can predict human judgments sufficiently accurately to serve as a baseline for future research. In the present work, rather than evaluating each of the individual features, we instead sought a means to (1) combine their strengths and (2) separate out both their shared and their complementary variance. We show that the space spanned by the features *en masse* is a useful quantitative tool for understanding human shape similarity. Indeed, we suggest that the precise feature set is less important than the space spanned by the features. Given the multiplicity of cells that contribute to representations of shapes and objects in ventral processing stream, it may not even be possible to describe a complete and unique set of features that the human visual system uses to describe shape. In fact, the response properties of cell populations may vary significantly across observers, yet similarity relationships between shapes could still be preserved. Hence it makes more sense to focus on the feature space as a whole, rather than the contributions of individual putative dimensions.

Another advantage of combining multiple features is the possibility to flexibly re-weight the features depending on the context or task. For example, Morgenstern, Schmidt, and Fleming [98] showed that in one-shot categorization observers tend to base their judgments of whether two novel objects belong to the same category on different features depending on the specific shapes to be compared. In a similar way, ShapeComp may explain context effects in shape similarity. For example, when *Shape A* is compared with *Shape B*, one feature may be more important in making up a similarity judgement than when *Shape A* is compared to *shape C*.

Although we did not explore this possibility here, feature re-weighting could also allow ShapeComp's high-dimensional space to resolve the tension between sensitivity and robustness to transformations. For example, where robustness to a particular transformation is important for a given task or judgment (e.g., rigid transformations for view-invariant object recognition) the visual system could increase the weight assigned to features that are least sensitive to that

transformation. For other tasks, where sensitivity to particular types of shape distortion are important (e.g., detecting subtle shape changes associated with the emotional state or intentions of an animal), the visual system could increase the gain associated with relevant features. Thus, multidimensional representations allow subsequent visual processes to selectively attend to different aspects of shape, optimizing features for task demands and environmental statistics [118, 119].

Because the features weights in ShapeComp are derived from the statistics of animal shapes, it is well suited to distinguishing natural shapes. It is intriguing that no fitting was necessary to predict human shape similarity judgments using ShapeComp—the raw weights derived from ca. 25,000 natural silhouettes account for most of the variance in **Fig 5B**. This suggests that natural shape statistics may play a central role in determining the space humans use to represent and compare shapes. What remains unclear, however, is (1) whether natural shape statistics bias shape similarity judgements in artificial shapes or (2) whether a high-dimensional shape space composed of any set of complementary shape features (even those optimized to differentiate artificial shapes) can predict human shape similarity. We have some support for (1): ShapeComp approximately predicts previous shape similarity data based on artificial stimuli. For example, Li et al. [90] constructed a 'perceptually circular' stimulus set, which ShapeComp predicts quite well (**Fig 9**). However, further work is needed to reveal the role of natural shape regularities in shape similarity perception.

Paired with a GAN trained on animal silhouettes, ShapeComp also provides a useful tool for automating the analysis and synthesis of complex naturalistic 2D shapes for future experiments in cognitive psychology and neuroscience. Novel, perceptually-uniform stimulus arrays can be generated and probed on the fly (**Figs 7** and **10**), for example, adaptively modifying stimuli in response to brain activity during an experiment. ShapeComp can also help create single- or multi-dimensional arrays (**Fig 10A–10C**), or stimulus sets that are perceptually equidistant from a given probe stimulus (**Fig 10D**). Once stimulus sets are controlled for image-based properties, the role of higher-level aspects of object representations can be probed in perception, visual search, memory, and other tasks.

## Limitations

There are a number of respects in which ShapeComp could be improved in further work. First, although humans can make many inferences from 2D contours (e.g., [13, 34, 35, 120]), for many applications it would be desirable to characterize similarity in 3D (e.g., computer vision and computer graphics [33]; video analysis [121]; topology mapping [122]; molecular biology [29], human tactile [123–125] and visual perception [126–128]). However, given that many of the 2D shape descriptors (**S1 Table**) have equivalents in 3D, and that ShapeComp is somewhat robust towards which descriptors are used in the model (**Fig 2H**), it is plausible that an implementation of ShapeComp based on 3D descriptors applied to 3D mesh representations would be a strong starting point for developing a model of human shape similarity in 3D.

Second, even highly reduced line drawings often provide additional cues for disambiguating form within the silhouette boundary [129–136]. For example, Pinna [137] showed how adding context such as inner line drawings could change our shape percepts as arising from one to two distinct objects. In addition, Wilder et al. [106] showed how symmetry within local contours of line drawings facilitates human scene categorization (See also: [138, 139]). Thus, there are many other ways to derive additional information from line drawings within a scene, in addition to the shape's silhouette, which are important for the coding of shape.

Third, shapes in the natural world are often occluded, while ShapeComp was trained only on non-occluded shapes. Occlusion is challenging because portions of the boundary of the

partially-hidden object are replaced with a completely different contour, belonging to the occluder. As ShapeComp is based on proximal shape features, rather than a deeper understanding of the distal causes of those features, it is ill-suited for comparing shapes across occlusion events. However, ShapeComp could serve as a benchmark to test the role of deeper scene understandings by characterizing the component of the judgments that can be explained purely by shallow image features in future research.

Fourth, ShapeComp was trained only on animal shapes. While the training set spans a very wide range of shape characteristics, future studies could refine ShapeComp by covering other major superordinate categories such as plants, furniture, tools and vehicles. This would probably modify the weighting of individual dimensions of ShapeComp, yet may further improve ShapeComp's predictions of human similarity judgments.

Fifth, while ShapeComp pools 109 different descriptors from across the literature, there are many others that were not included. Incorporating additional features would likely change the precise estimates of similarity made by ShapeComp (although, **Fig 2H** suggests that using different subsets of features yields similar composite dimensions in MDS). Yet, we believe that there is no one single shape descriptor that perfectly captures all of human shape similarity perception, and that the general approach of pooling multiple descriptors provides robust and sensitive representations.

Sixth, as a model of human perception, ShapeComp is entirely parameter-free in the sense that no fitting was used to adjust the features or their weights to improve predictions of human judgments. We saw this as an important component of testing whether weightings derived from natural shapes predict human perception. However, with over 100 features, ShapeComp's predictions could almost certainly be further improved by explicitly fitting to human data. However, as noted above, in the human visual system the weighting of features may even adjust flexibly depending on context or task [e.g., 99]. In future work, it would be interesting to test whether adding bottom-up or top-down gain control pathways to dynamically regulate features weights, better captures the effects of context-sensitive normalization and attentional control in human shape similarity judgments.

Finally, ShapeComp is not a physiologically plausible model of shape representation processes in the human brain. Future research should seek to model in detail the classes of features in the neural processing hierarchy that represent shapes in a multidimensional space [140]. We believe that paired with novel image-generating methods, like GANs, ShapeComp can play a central role in mapping out visual shape representations in cortex.

## Conclusions

Shape can be described in many different ways, which have complementary strengths and weaknesses. We have shown that human shape similarity judgments can be well predicted by combining many different shape descriptors into a multidimensional representation. The ShapeComp model correctly predicts human shape perception across a wide range of conditions. It captures perceptual subtleties that conventional pixel-based metrics cannot, and provides a powerful tool for generating and analysing stimuli. Thus, ShapeComp not only provides a benchmark for future work on object perception, but also provides a proof-of-principle account of how human shape processing is simultaneously sensitive, robust and flexible.

## Methods

### Ethics statement

All procedures were approved by the local ethics committee of the Department of Psychology and Sports Sciences of the Justus-Liebig University Giessen (Lokale Ethik-Kommission des

Fachbereichs 06, LEK-FB06; application number: 2018–0003) and adhered to the declaration of Helsinki. All participants provided written informed consent prior to participating.

### Sensitivity/robustness analysis to transformation

**Shape descriptors.** Shape descriptors consisted of simple descriptors like area and perimeter, to more complex descriptors like the shape skeleton. A full list of the 109 descriptors is listed in **S1 Table**.

**Transformation analysis.** We illustrate the complementary nature of different shape descriptors by transforming one sample from each of 20 animal categories (e.g., birds, cows, horses, tortoise; from [141, 142]) with four 2D transformations (rotation, shear, 'bloating' and noise) of varying strengths. More specifically, the transformations applied to the $x,y$ coordinates of each shapes were as follows:

A)  **Rotation**: We use a rotation matrix $R = \begin{bmatrix} \cos\theta & -\sin\theta \\ \sin\theta & \cos\theta \end{bmatrix}$ to rotate the shape around its centroid such that new$[x,y] = R \times$ shape$[x,y]$.
    We produced 23 new variants by sampling $\theta$ every 15˚.

B)  **Shear**: we applied a shear transform $S = \begin{bmatrix} 1 & 0 \\ a & 1 \end{bmatrix}$ that slants the shape along the y-axis by factor $a$ such that new$[x,y] = S \times$ shape$[x,y]$.
    We used 5 different levels of $a$ ranging from 0.2 to 1.

C)  **Bloating**: we 'bloat' the shape with the following transform, such that:

$$\text{new}[x, y] = \text{shape}[cartx(r^{0.75}, \ \theta), carty(r^{0.75}, \ \theta)].$$

where $r$ and $\theta$ give the radius and angle of location $x$ and $y$ from the shape centroid, and *cartx* and *carty* convert from polar to Cartesian coordinates. We created bloats of increasing magnitudes by iteratively passing a shape through the transformation up to 4 times.

D)  **Noise**: we add random Gaussian noise $N(0, \sigma)$ to shape's $x,y$ position. Noise levels varied from small (0.5% of the maximum distance between any two contour points in a given shape) to large (4% of the max distance), such that new$[x,y] =$ shape$[x,y] + N(0, \sigma)$.

For each animal category and shape descriptor, we compute the sensitivity of a given transform (e.g., rotation or bloating), $S_{ij}$, where $i$ represents 1 of 20 animal categories, and $j$ one of the 109 shape descriptors. Specifically, we examined how sensitive each shape descriptor $j$ was to a given transformation by computing the mean differences between shape descriptors for the original shape with the transformed version, as follows:

$$S_{ij} = \frac{\sum\limits_{t=1}^{n} \dfrac{\sqrt{(s_{oj} - s_{tj})^2} - d_{\min}}{d_{\max} - d_{\min}}}{n}$$

where $s_{oj}$ is shape descriptor value for the original shape, and $s_{tj}$ is descriptor value on one of the $n$ transformed versions. Given different descriptors are in different units and thus show a different range of values, to compare sensitivity across descriptors and transformations, we normalize the differences between the original and transformed shape descriptor with $d_{\min}$ and $d_{\max}$, where $d_{\min}$ is the smallest difference between shape $s_{oj}$ and any of its transformed versions $s_{tj}$ (including across other comparison transformations), and $d_{\max}$ is the largest

difference between $s_{oj}$ and any of its transformed version (also including across other comparison transforms like rotation, bloating or noise). Larger values of $S_{ij}$ indicate that the descriptor is sensitive to the transformation (i.e., the transformation has a stronger influence on the shape descriptor). Using MATLAB function 'nanmean' to ignore taking the mean across undefined results (e.g., 0/0 or 0×Inf), we then took the mean across the 20 samples as the sensitivity of the shape descriptor to a given transformation, where larger values indicate more sensitivity:

$$S_{Tj} = \frac{\sum_{i}^{N=20} S_{ij}}{N},$$ where N is 20 the number of animal categories.

## Real-world shape analysis

**Animal shape analysis.** We amassed 25,712 animal shapes—purchased from shutterstock (e.g., Natalia Toropova; Big animal silhouttes set), based on 3D animal models (purchased from https://evermotion.org; e.g., archmodels volume 83) or gathered from previous work (e.g., [141, 142]). The 3D animal mesh models were used to render a number of additional 2D silhouettes with varying elevation and azimuth angles. Together, these >25,000 shapes came from many different animal categories with the bulk being mammals (e.g., dogs, cats, apes, horses), but also including other categories like fish, reptiles, or insects. For each animal shape, we calculated 109 shape descriptors (listed in **S1 Table**) thought to be important for recognition, synthesis, and perception [32]. The shapes' *x,y* coordinates (384×2 resolution) were sampled uniformly and scaled to {0–1} by first subtracting the absolute minimum value of each coordinate, and then diving by the resulting absolute maximum value. Twenty-six of the shape descriptors (e.g., shape context) were computed along the contour and require an initial point for shape matching. Rather than using a matching strategy that depends on context and thus would differ as one shape is compared to another, we chose a strategy that would be the same across all shapes, that is, to set the point with the smallest x-value (i.e., leftmost point). In cases when the smallest x-value on a contour repeated–for example, when it reappeared in a neighboring point (~ 3.5% of animal shapes in the database), or a point further along the contour (~0.3% of shapes in the database), we chose randomly among the repeated points as the initial shape point.

**Multidimensional scaling and ShapeComp model.** We used classical MDS to find an orthogonal set of shape dimensions that captures the variance in the animal dataset. Specifically, for each shape descriptor, we computed the Euclidean distance between each pair of shapes in the dataset:

$$d^{ij} = \sqrt{\left(f_k^i - f_k^j\right)^2} = \sqrt{\left(\Delta f_k^{ij}\right)^2}$$

Where $d^{ij}$ is the distance between stimulus *i* and *j* on shape descriptor *k* and $f_k^i$ and $f_k^j$ are the values on shape descriptor *k* for stimuli *i* and j. Once the computation for all pairwise comparisons was complete, the distances were assembled into a 25,712 × 25,712 similarity matrix and normalized by their largest distance. We computed this normalized distance, $\hat{d}$, for all shapes and shape descriptors to form a 25,712 × 25,712 × 109 entry matrix (shapes$^2$ × shape descriptors). We then computed a 109-dimensional Euclidean distance *D* across the shape descriptors for shape pair *i* and *j*, as follows:

$$D^{ij} = \sqrt{\sum_{k=1}^{109} \left(\hat{d}_k^{ij}\right)^2}.$$

We then computed classical MDS on the resultant 25,712 × 25,712 similarity matrix, taking the first 22-dimensions (see **Fig 2** and **Results:** *Analysis of real-world shapes*) as the Shape-Comp model.

**Comparison ShapeComp spaces.** To test the robustness of the ShapeComp model's high-dimensional space, we compare the spaces computed across different (1) animals shapes and (2) combination of features. In (1) we selected 10 groups of 500 different animal shapes. We computed a separate ShapeComp space for each of the 10 groups (as described in the preceding section but with 500 samples instead of >25,000 samples). In (2) we computed a separate ShapeComp space for the same 500 animal shapes, but with a random combination of 55 out of the 109 shape descriptors. To compare the consistency across the spaces in (1) and (2), we created a test set with 200 test shapes that were not included in creating any spaces. We then moved the 200 test shapes into each new ShapeComp space (see **Methods:** Estimating coordinates for new shapes in pre-existing shape spaces). For each shape space, we then computed the pairwise distances across the 22 dimensions for each test shape yielding a 200 x 200 similarity matrix. We then computed the Pearson correlation of the upper triangular matrix of each similarity matrix across the different spaces as a test of ShapeComp's robustness.

**Estimating coordinates for new shapes in pre-existing shape spaces.** We estimate the coordinates for a new shape in the high-dimensional animal MDS space by (1) comparing the shape descriptors for the new shape with a subset of >25,000 animal shapes, (2) computing a new MDS solution, and then (3) using Procrustes to move this new MDS solution to the high-dimensional animal MDS space. Specifically, we computed the Euclidean distance between the new shape and 500 shapes already located in the animal space, to assemble a 501×501 similarity matrix, and scaled by the largest distance for each feature distance in the complete animal dataset. We did this for all shape descriptors to form a 501×501×109 matrix (shapes$^2$ × shape descriptors). We then computed the 109-dimensional Euclidean distance $D$ across shape descriptors yielding a 501×501 similarity matrix. Applying Classical MDS produced a new coordinate space for the original 500 shapes. We used Procrustes analysis to identify the linear transform that maps the MDS coordinates for the 500 animal shapes from the new coordinate space to the original coordinate space. We then applied this transformation to the new shape to move it into the original shape space.

## Perception of real-world shapes

**Participants and stimuli.** *15* participants (mean age: 24.7 years; range 20–35) arranged two sets of twenty shapes (rabbits and horses) from Bai et al. [141, 142]. *10* different participants (mean age: 30.4; range 25–39) arranged 1 set of 30 shapes that varied across 5 animal categories (i.e., spiders, turtles, rabbits, horses, and elephants). All participants were paid 8 Euros per hour, and signed an informed consent approved by the ethics board at Justus-Liebig-University Giessen and in accordance with the Code of Ethics of the World Medical Association (Declaration of Helsinki). Participants reported normal or corrected-to-normal vision.

**Procedure.** All experiments were run with an Eizo ColorEdge CG277 LCD monitor (68 cm display size; 1920 × 1200 resolution) on a Mac Mini 2012 2.3 GHz Intel Core i7 with the psychophysics toolbox [143, 144] in MATLAB version 2015a. Observers sat 57cm from the monitor such that 1 cm on screen subtended 1˚ visual angle.

Experiments were run in MATLAB using the multi-arrangement code provided by Krieges-korte & Mur [49] and adapted for the Psychophysics Toolbox. On each trial, participants used the mouse to arrange all stimuli by their similarity relationships to one another within a circular arena. At the start of each trial, stimuli were arranged at regular angular intervals in random order around the arena. To the right of the arena, the current and last selected objects were

shown larger in size (15˚). Once an arrangement was complete, participants pressed the Return key to proceed to the next trial. The next trials showed a subset of the objects from the first trial based on the '*lift-the-weakest*' algorithm [49]. The arrangements ended after 12 minutes had elapsed.

## GAN shapes

GANs are unsupervised machine learning systems that pit two neural networks against each other [145, 146] (**Fig 3A**), The GAN was trained using MatConvNet in MATLAB to synthesize shapes that it could not distinguish from the animals shapes database. The network architecture and hyperparameters were the same as in Radford et al. [146], except for the following. The latent *z* vector was 25×1 (rather than 100×1) and one of the dimensions of the remaining filter sizes was reduced (from initially matching the other dimension) to 2. A series of four "fractionally-strided" convolutions then converted the latent vector's high-level representation into the shapes' spatial coordinates. We generated novel shapes using the generator network trained after 106 epochs by inputting random vectors into its latent variable. We blurred the shapes with a Gaussian filter with a standard deviation of two neighbouring contour points and selected shapes without self-intersections.

**Visualizing ShapeComp dimensions.** To aid interpretation of ShapeComp, we sought to visualize which shape qualities each dimension independently describes. Accordingly, for each dimension of ShapeComp, we sought shapes that varied along that dimension, while minimizing the variations along the other dimensions. To create such shapes, we used the Genetic Algorithm (GA) in MATLAB's Global Optimization toolbox, in combination with a neural network (see *ShapeComp Network* and **Fig 7**) that takes as input a shape and returns as output the shape's coordinates in ShapeComp's 22-dimensional space. Specifically, with a population of 200 neural networks for 250 generations, the objective of the GA was to find shapes in GAN space that varied along one dimension in ShapeComp while the remaining dimensions are held roughly constant. The ShapeComp network, its architecture, and error in predicting ShapeComp are described in more detail in **Methods:** *Shape to ShapeComp Network*.

## GAN vs. animal shapes category judgement experiment

**Participants.** In total, there were forty participants (mean age: 24.4 years; range 19–33). Half of the participants classified GAN shapes, and the other half classified animal shapes

**Stimuli.** Photographs (9×12.5 cm) of 100 GAN shapes with no-self intersections (randomly selected from the GAN latent space) and 20 animal shapes from Bai et. al [141, 142]. Each photograph had a number to indicate shape (1–100 for GAN shapes, 1–20 animal shapes).

**Procedure.** Experimenter shuffled the cards, and placed them in front of participant. Participant picked up the top card and placed it roughly arm's length from their view. They called out the number on the card, and were then asked to judge the category of the shape on the card. Participants had the option of saying that the shape does not appear like any known category. Experimenter entered the responses, while the participant picked up the next card from the pile. This process continued until the participant finished classifying the whole stack.

## Shape similarity rating experiment

**Participants.** 14 observers participated in the shape similarity rating experiments. Mean age was 24.4 (range: 21–33). Participants, paid at a rate of 8 euros per hour, signed an informed consent form approved by the ethics board at Justus-Liebig-University Giessen and in

accordance with the Code of Ethics of the World Medical Association (Declaration of Helsinki). Participants reported normal or corrected-to-normal vision.

**Procedure.** As in the other experiments, the experiments were run with an Eizo Color-Edge CG277 LCD monitor (68 cm display size; 1920 x 1200 resolution) on a Mac Mini 2012 2.3 GHz Intel Core i7 with the psychophysics toolbox [124, 125] in MATLAB version 2015a. Observers sat 57cm from the monitor such that 1 cm on screen subtended 1˚ visual angle.

**Pairwise similarity ratings.** 250 GAN shape pairs were chosen that spanned a large range of distances in ShapeComp. On each trial, stimuli were shown side by side and observer adjusted a slider to indicate similarity ratings from 0 ('very dissimilar') -100 ('very similar') using the mouse. Shapes subtended ~15˚. Shape position (right or left side) was randomized on each trial. Shape pairs were presented in random order.

## Shape Triads Judgements: Pixel similarity and ShapeComp dimensions experiments

**Participants.** 19 different observers participated in the *pixel similarity* and *ShapeComp dimensions* experiment. Mean age was 24.3 (range: 20–33). Participants, paid at a rate of 8 euros per hour, signed an informed consent approved by the ethics board at Justus-Liebig-University Giessen and in accordance with the Code of Ethics of the World Medical Association (Declaration of Helsinki). Participants reported normal or corrected-to-normal vision.

**Procedure.** We used the same setup as described in **Methods:** *Shape similarity rating experiment.*

**Pixel similarity triplets.** Stimuli were created using the GAN trained on animal silhouettes (see **Results:** *Using Generative Adversarial Networks to create novel naturalistic outlines*). Using the Genetic Algorithm in MATLAB's Global Optimization toolbox with a population of 200 neural networks for 250 generations, we used the ShapeComp network (described in **Methods:** *Shape to ShapeComp Network*) to find triplets of GAN shapes in which a *sample* shape varied in its ShapeComp distance from two *test* shapes, $t_A$ and $t_B$, while maintaining the same pixel similarity to both. Specifically, we computed the ShapeComp distance from the *sample* to each *test*, $a$ for $t_A$ and $b$ for $t_B$ (**Fig 5E**). We then represented the distances from these test shapes to the sample as a ratio between the smaller of the distances to the sum of their distances:

$$min(a,\ b)/(a+b)$$

Small values of this ratio indicate one *test* stimulus was much closer to the *sample* shape than the other, in terms of ShapeComp. A maximum value of 0.5 indicates both tests are equally far from the sample. 70 triplets were created and binned into 7 bins ranging 0.2–0.5, where each bin contained ~10 triplets. On each trial, the *sample* shape was presented centrally, flanked by two *test* shapes (whose position, left or right of *sample* was randomized). Shapes subtended 12˚. Pixel similarity, held constant between the *sample* and the *test* shapes, was defined as the Jaccard index (1—intersection-over-union; [37]). High values indicate high pixel similarity.

**ShapeComp dimensions triplets.** Similar to the pixel similarity experiment, using the Genetic Algorithm in MATLAB's Global Optimization toolbox with a population of 200 neural networks for 250 generations, we used the ShapeComp network (described in **Methods:** *Shape to ShapeComp Network*) to find shape triplets in which a *sample* shape varied in its ShapeComp distance from two *test* shapes, $t_A$ and $t_B$, while maintaining the same value on one of the ShapeComp dimensions {1–8}. The distance between *sample* and *test* shapes was represented with the ratio described in *pixel similarity triplets*.

### Identifying perceptual nonlinearities in shape spaces of novel objects

**Procedure.**   Experiments were run in MATLAB using the multi-arrangement code provided by Kriegeskorte & Mur [49]. The procedure was the same as in **Methods:** *Perception of real-world shapes*.

**Participants.**   Two groups of 16 observers (mean age: 24.45 years; range: 18–41), including the first author who was the only author and participant in both groups.

**Stimuli.**   Four GAN shape sets were selected that ranged in their correlation with Shape-Comp22 network ($0.56 \leq r \leq 0.74$). One group of participants arranged two sets with 20 shapes (set a, $r = 0.56$; set b, $r = 0.69$). Another group arranged two sets with 25 shapes (set c, $r = 0.74$; set d; $r = 0.72$).

### Deriving perceptually uniform shape spaces of novel objects

**Procedure.**   Experiments were run in MATLAB using the multi-arrangement code provided by Kriegeskorte & Mur [49]. The procedure was the same as in **Methods:** *Perception of real-world shapes*.

**Participants.**   Two groups of 16 observers (mean age: 25.03 years; range: 18–41), including the first author who was the only author and participant in both groups.

**Stimuli.**   Four sets of 25 shapes for which the GAN's latent vector and the ShapeComp neural network (described in more detail in **Methods:** *Shape to ShapeComp Network*) predicted similar pairwise distances ($r > 0.9$). One group of participants arranged two shape sets that were uniform in ShapeComp (set A and B). Another group arranged two shape sets that were uniform in GAN space (set C and D).

### Shape to ShapeComp network

We trained several instances of a convolutional neural network, one in MATLAB's neural network toolbox and two in Keras with TensorFlow–an open source neural network library in Python. The networks were trained to take as input a 384×2 contour or 40×40 image patch through multiple neural layers (shown in **Fig 7A–7C**) and output the 22-dimensional MDS coordinate. To do this, we created a set of 950,000 GAN shapes (800,000 training, 150,000 test images) and then computed their 22D ShapeComp coordinates (see **Methods:** *Estimating coordinates for new shapes in pre-existing shape spaces* described above*)*. These coordinates served as the desired network output. The network architecture and training hyperparameters are shown in **Fig 7A–7C**. Input shapes yield an estimate of the 22D ShapeComp coordinate as output. We used the MATLAB neural network implementation to visualize the ShapeComp dimensions in **Fig 4** and to select the stimuli in Experiments with shape-triads and shape spaces (described above). The purpose of the additional Python-based networks was to provide cross-platform capabilities.

The MATLAB (**MatNet**), and one of the Keras (**KerNet1**) networks assume that the minimum *x*-value of the contour is the first point along the shape, as this is how shape descriptors that required an initial starting point in the animal database were calculated. While the contour representation is efficient—it packs much more detail about a shape given the same amount of information (in terms of bytes) in an image representation—it also has shortcomings. One major limitation is the correspondence problem associated with matching a point on one shape with another. Here we used a simple heuristic—setting the left most point as the first point on any contour. However, this rule has its own shortcomings. Imagine, for example, rotating a shape with multiple limbs. As the shape rotates even by a minor amount, the left-most point can quickly shift a large number of points as a new limb transitions into this position making the first point selected across the two similar shapes (and thus their ShapeComp distances) potentially highly different. One solution is to use the image of the shape rather

than its contour, as this would bypass the correspondence problem. Moving in this direction, the second Keras network (**KerNet2**) was trained to compute ShapeComp using a 40×40 pixel image as input, rather than a contour represented as (x, y) coordinates.

### CNN Shape similarity

We evaluated pre-trained CNNs with MATLAB's neural network toolbox. Shapes were converted to images that matched the input size of the network. The images showed the shapes with RGB values of 0 and background with values of 255, however changing the way the shapes were coded (e.g., inverting the RGB relationships) did not bring large changes in the results. Following Kubilius et al. [91], we defined network shape similarity as Euclidean distance in their final fully-connected layer (with 1000 units).

## Supporting information

**S1 Fig. Over 100 shape descriptors evaluated in terms of their 'sensitivity', i.e., how much they changed when shapes were transformed by noise and shear.** Here, solidity, area, and curviness are more sensitive to noise than shear, while major axis orientation is less sensitive to noise than shear. That different descriptors are tuned to different transformations highlights their complementary nature.
(TIFF)

**S2 Fig. The original features that best account for ShapeComp.** A wordcloud that shows the 20 best features in terms of absolute correlation to each of ShapeComp's first 8 dimensions (A-H) and (I) across all 22-dimensions. The largest words in the cloud, the most predictive features, are highlighted with colour.
(TIFF)

**S3 Fig. The original features that account least for ShapeComp.** A wordcloud that shows the 20 features that least predictive (in terms of absolute correlation) to each of ShapeComp's first 8 dimensions (A-H) and (I) across all 22-dimensions. The largest words in the cloud, the least predictive features, are highlighted with colour.
(TIFF)

**S1 Table. List of 109 shape descriptors in ShapeComp.**
(DOCX)

## Acknowledgments

We thank Saskia Honnefeller, Jasmin Kleis, and Marcel Schepko for their help setting up the experiments and running initial pilot studies.

## Author Contributions

**Conceptualization:** Yaniv Morgenstern, Frieder Hartmann, Filipp Schmidt, Henning Tiedemann, Eugen Prokott, Guido Maiello, Roland W. Fleming.

**Data curation:** Yaniv Morgenstern.

**Formal analysis:** Yaniv Morgenstern.

**Funding acquisition:** Guido Maiello, Roland W. Fleming.

**Investigation:** Yaniv Morgenstern, Roland W. Fleming.

**Methodology:** Yaniv Morgenstern, Frieder Hartmann, Filipp Schmidt, Eugen Prokott, Roland W. Fleming.

**Project administration:** Yaniv Morgenstern, Roland W. Fleming.

**Resources:** Roland W. Fleming.

**Software:** Yaniv Morgenstern, Frieder Hartmann, Roland W. Fleming.

**Supervision:** Roland W. Fleming.

**Validation:** Yaniv Morgenstern.

**Visualization:** Yaniv Morgenstern, Filipp Schmidt, Roland W. Fleming.

**Writing – original draft:** Yaniv Morgenstern, Roland W. Fleming.

**Writing – review & editing:** Yaniv Morgenstern, Frieder Hartmann, Filipp Schmidt, Henning Tiedemann, Eugen Prokott, Guido Maiello, Roland W. Fleming.

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
