## [Decision Letter · Decision Letter 0]

1 Dec 2020

Dear Dr Morgenstern,

Thank you very much for submitting your manuscript "An image-computable model of human visual shape similarity" for consideration at PLOS Computational Biology.

Your manuscript was reviewed by members of the editorial board and by three independent reviewers. As you will notice in the reviews (below this email), the Reviewers are generally positive about the model, but they have mixed opinions about the ultimate contribution of this study. While Reviewers #1 and #2 believe that the study "will be of interest to many scholars in the field" and "represents an important advancement in the field", respectively, Reviewer #3 states that "the conclusions drawn in the manuscript [are] restatements of things that we already know".

In light of the reviews, we would like to invite the resubmission of a significantly-revised version that takes into account the reviewers' comments. It will be of particular importance to spell out more clearly the study's contribution to the field, viewed from the perspective of the readership of PLOS Computational Biology.

We cannot make any decision about publication until we have seen the revised manuscript and your response to the reviewers' comments. Your revised manuscript is also likely to be sent to reviewers for further evaluation.

Sincerely,

Ronald van den Berg

Associate Editor

PLOS Computational Biology

Wolfgang Einhäuser

Deputy Editor

PLOS Computational Biology

Reviewer's Responses to Questions

**Comments to the Authors:**

Reviewer #1: The authors define a model that combines information from many different shape representations and show how it can be used to explain human shape similarity judgments. Their model is very comprehensive and represents an important advancement in the field. Other recent attempts to define a shape space have been to simple and too constrained to small sets of images and specific tasks to be relevant for understanding the full complexity of the shapes of our world and the wide variety of tasks that involve interacting with them.

Overall, the authors do a good job of describing the use of the model, and referring the reader to the components of the model.

The only part I find lacking is any discussion about dimensionality reduction. The authors use MDS to reduce the number of dimensions from 109 to 22. These 22 dimensions are combinations of the original 109. I would have liked to know more about the usefulness of many of the original 109 dimensions. Are there any that are not worth including? Can any be removed without affecting the 22 dimensional-space that gets recovered? Which of the original 109 dimensions load most strongly onto the 22 dimension? Can anything be said about which of the original dimensions are most important to include?

Following this line of thinking I was wondering if the authors tried to use the CNN to recover the original 109 dimensions, or only the 22 dimensions. Beyond having the reduced dimensions, is there something about the 22 dimensional space that makes it easier to accurately determine the ShapeComp representation? Or would it be just as easy to compute the 109 dimensions and use the same transformation to reduce it to the 22 dimensions?

As computing the 109 dimensions using a collection of shape processing tools would be extremely burdensome, the inclusion of the CNNs that can give similar enough results with a single tool is very helpful. The authors do not explain how to make use of each component, so it would be difficult to reproduce this work, and would require looking back at many different articles, however, while I would normally find this problematic, but releasing a CNN that can do something similar allows for the community to benefit from the authors’ work.

Overall, I find these results important and useful for the field. This work was an immense undertaking and the field will benefit from its publication.

Reviewer #2: The manuscript describes a model which uses 109 shape descriptors from the scientific literature to predict human shape similarity judgments between pairs of shapes. The model has a number of significant strengths (documented throughout the manuscript) and, of course, some weaknesses (kudos to the authors for nicely describing these weaknesses toward the end of the manuscript).

I have mixed feelings about this manuscript but, ultimately, I believe its negative features outweigh its positive features.

In brief, the manuscript attempts to advance our understanding of human visual shape perception. However, it never tells the reader the new and important insights provided by this research. Indeed, I found the conclusions drawn in the manuscript to be restatements of things that we already know. Consequently, I don't feel as if I learned anything new on the basis of this research.

For example, a major conclusion of the manuscript is stated as follows (page 7): "More generally, the plot shows the wide range of sensitivities across different shape metrics, indicating that depending on the context or goal, different shape features may be more or less appropriate." I agree completely. But I (and nearly all other researchers in the field) knew this already. Does the reported research shed any new light?

As a second example, the manuscript concludes (page 10): "Thus, while not all 109 shape descriptors are independent, a multidimensional space is indeed required to capture the variability inherent in animal shapes." Again, I agree completely. But, again, everyone in the field already knew this. I've never yet met a researcher in the field that thought that visual shape is simple, so simple that a one-dimensional space suffices to capture the variability inherent in animal shapes. Again, it seems important to ask if the research reported here sheds any new light?

As a third example, the manuscript concludes (page 19): "Consistent with previous works [44, 79-85, 87], this confirms that human shape similarity relies on more sophisticated features than pixel similarity alone." Yes, of course. Everyone in the field already believes this. Was there some doubt in the field, thus requiring further investigation?

I could go on and on. Here are a couple of more quotes from the manuscript. On page 20, the manuscript states, "This indicates that human shape perception relies on more than a single ShapeComp dimension." On page 20, the manuscript states, "Together, these results show that human shape similarity relies on multiple ShapeComp dimensions--highlighting the importance of combining many complementary shape descriptors into ShapeComp." Yes, of course. I agree with all these statements and many more too. So does everyone else in the field. I wish the manuscript told us the new and important insights provided by the authors' research.

(As an aside, if it seems as if I'm repeating myself here, it is because the manuscript is very repetitive. I estimate that it could be cut by 40%-50% without loss of meaningful content.)

Here are a few other comments that may be helpful to the authors:

The authors should keep in mind that shape similarity is generally thought of as a means toward an end, not as an end in itself. For instance, shape similarity estimates might be useful in a system that performs visual object recognition, a system that plans motor movements for grasping, a system that performs problem solving and action planning, etc. I encourage the authors to use their model for one of these applications, and then write a manuscript about the great performance of their system (relative to other systems).

The manuscript mentions one or two applications of the model toward its end. For instance, the manuscript shows how the model can be used to derive perceptually uniform shape spaces of novel objects. I agree that this might be useful for experimentalists in the vision sciences. However, without careful comparisons, there is no way of knowing whether the method described here is better or worse than its alternatives.

Lastly, the manuscript states that a limitation of the proposed model is that it only considers 2D shape descriptors, whereas "for many applications it would be desirable to characterize similarity in 3D" (page 34). I fully agree. To me, this seems like a great area that is ripe for new and important insights.

Reviewer #3: In this manuscript, the authors developed (and tested) a model to capture the human ability to perceive shape similarities. Overall, the results are very promising and suggest that human ability to perceive shape similarities can be explained by combining a large number of shape dimensions and that multiple dimensions better perform over and above every single dimension or shape silhouette. The model (‘ShapeComp’) reaches human-level performance with real-world object shapes (animal shapes) but also when using novel shapes, and across a wide range of tasks (e.g., similarity judgments across pairs of shapes, multiple object similarities). The results appear very promising despite some limitations, partially already discussed in the manuscript.

The study tackles a relevant problem across several research fields from computer vision to cognitive neuroscience and goes one step further by combining a large number of shape descriptors in trying to capture the human (special) ability to perceive the shape. Many are the models that have tried to address this question mainly focusing on one or two dimensions. Together, these results further support the multidimensional nature of human perception and highlight the limitations of focusing on one/few single dimensions to describe shape perception. This study will be of interest to many scholars in several fields.

I enjoyed reading this manuscript and I appreciated particularly the critical approach taken by the authors. While reading this work, I posed myself several questions, which happen to be addressed one by one in the subsequent sections of the manuscript. For instance, based on Figure 2A, I asked myself whether much simpler models such as a simple shape silhouette would be enough to capture most variance in the data; I was happy to see that the authors also considered this control analysis. I also appreciated the subsequent controls to rule out the contribution of individual descriptors. In addition to this, I feel it would be relevant to test the ShapeComp against the performance of recent deep neural network. For instance, it has shown that pre-trained DNNs are good models to approximate human ability to perceive shape (Kubilius et al., 2016). It would be particularly relevant to test to what extent the ShapeComp model can better explain human shape perception relative to standard pre-trained DNNs.

The authors used real-word shapes of animals (> 25000) as input information. I appreciate this choice. As the authors also mention, this specific choice was driven by the consideration that human vision is exposed (since birth) to real-world shapes of meaningful objects. Therefore, a model of human shape needs to account for those perceptual biases driven by high-level aspects such as object meaning, class, or context that humans will probably try to extrapolate even when confronted with novel shapes. For this reason, I think it would be useful to test whether this model can capture intrinsic biases that might shape human shape perception. For instance, if I understand correctly from the methods section, in the analysis reported in figure 2B and 2C the authors tested the ability of ShapeComp to capture human similarity judgments in two sets of shapes within the same animal class (houses and rabbits). It would be interesting to test whether this result generalises to subsets of images that span a few animal classes. Would humans use the same strategy in both situations: when asked to arrange shapes for objects (e.g., animals) within the same class vs multiple classes? Or would the latter setting result in a shape arrangement that does also take into account high-level object information? Since human perception does not happen out of context, this test would be relevant when trying to approximate human shape perception.

For the same reason mentioned above, meaning that, perception does not happen in a vacuum, I feel that reducing the input to shape outline only, might be limiting the potentiality of such a model. I wonder whether including additional descriptors that capture information from the object line drawing, which provide important shape features, could better capture more realistic human shape perception abilities. Two different objects might have the same outline when seen from a specific viewpoint but could nevertheless be distinguishable if depicted with line drawing. To be honest, the authors already discuss some of the model’s limitations in similar directions (e.g., 3D object information). However, I would appreciate a further discussion of aspects that take into consideration real object perception.

It is interesting to see that when combining multiple shape descriptors the model can reach good performance regardless of what subset of dimensions is considered. I agree with the authors that this should not be taken as evidence for equal importance/contribution of the descriptors but shows that the descriptors inevitably overlap to a certain degree and if many descriptors are considered it becomes less relevant what specific subset is selected. I wonder whether this might also be a consequence of the input choice, as mentioned above; using object outlines reduces the available information, possibly resulting in the different descriptors being more correlated among each other. This is not a critique, is just a thought. Am curious to know the authors’ thoughts on this point.

**Have all data underlying the figures and results presented in the manuscript been provided?**

Reviewer #1: Yes

Reviewer #2: None

Reviewer #3: Yes

PLOS authors have the option to publish the peer review history of their article (what does this mean?). If published, this will include your full peer review and any attached files.

Reviewer #1: No

Reviewer #2: No

Reviewer #3: No
---

## [Decision Letter · Decision Letter 1]

19 Apr 2021

Dear Dr Morgenstern,

We are pleased to inform you that your manuscript 'An image-computable model of human visual shape similarity' has been provisionally accepted for publication in PLOS Computational Biology.

Best regards,

Ronald van den Berg

Associate Editor

PLOS Computational Biology

Wolfgang Einhäuser

Deputy Editor

PLOS Computational Biology

Reviewer's Responses to Questions

**Comments to the Authors:**

Reviewer #1: In this revision the authors added additional analyses related to dimensionality reduction, they clarified their motivation and claims of the paper, and they added an additional experiment.

In these modifications the authors adequately responded to all of my comments and to those of the other reviewers.

Reviewer #2: The revised manuscript is much improved over the original. I thank the authors for taking seriously the reviewers' comments. This revision makes it clear that the authors' contribution is largely an engineering one -- whereas previous researchers have developed a large number of individual shape descriptors, the current authors have designed and implemented a system that combines multiple shape descriptors from the literature. While it is not surprising that this composite system outperforms simpler systems, the manuscript does a good job of documenting the proposed system's strengths and weaknesses. I now think that this manuscript will make a worthwhile contribution to the literature.

Reviewer #3: I appreciate the authors' consideration and effort to address the reviewers' comments.

I don't have any further comment.

**Have the authors made all data and (if applicable) computational code underlying the findings in their manuscript fully available?**

Reviewer #2: **No: **The manuscript states that code will be made available if the manuscript is accepted for publication.

Reviewer #3: Yes

PLOS authors have the option to publish the peer review history of their article (what does this mean?). If published, this will include your full peer review and any attached files.

Reviewer #1: No

Reviewer #2: No

Reviewer #3: No

**Have all data underlying the figures and results presented in the manuscript been provided?**

Reviewer #1: Yes

---

## [Editor Report · Acceptance letter]

26 May 2021

PCOMPBIOL-D-20-01749R1 

An image-computable model of human visual shape similarity

Dear Dr Morgenstern,

I am pleased to inform you that your manuscript has been formally accepted for publication in PLOS Computational Biology. Your manuscript is now with our production department and you will be notified of the publication date in due course.

With kind regards,

Agota Szep
